# Epithelial–Macrophage Crosstalk in Host Responses to *Campylobacter jejuni* Infection in Humans

**DOI:** 10.3390/microorganisms13122808

**Published:** 2025-12-10

**Authors:** Khaled Abdelaziz, Shreeya Sharma, Mostafa Naguib, Alexis Stamatikos

**Affiliations:** 1Department of Animal and Veterinary Science, Clemson University, Clemson, SC 29634, USA; shreeys@g.clemson.edu (S.S.); mnaguib@g.clemson.edu (M.N.); 2Clemson University School of Health Research (CUSHR), Clemson, SC 29634, USA; 3Department of Poultry Diseases, Faculty of Veterinary Medicine, Cairo University, Cairo 12211, Egypt; 4Department of Food, Nutrition, and Packaging Sciences, Clemson University, Clemson, SC 29634, USA; adstama@clemson.edu

**Keywords:** macrophage, epithelial cell, intestine, immunity, *Campylobacter*, communication, crosstalk

## Abstract

Interactions between *Campylobacter jejuni* and host immune cells have been studied using various single-cell line models, such as macrophages and intestinal epithelial cells; however, these single-cell approaches do not fully capture the complexity of the host response. Investigating the interactions between these cell types offers a more comprehensive model for understanding *Campylobacter*–host dynamics. Therefore, this study aimed to investigate these interactions, specifically between intestinal epithelial cells and macrophages, using an in vitro model of *C. jejuni* infection. We examined whether soluble factors secreted from *C. jejuni*-infected HT-29 cells (human colorectal adenocarcinoma cells that express characteristics of mature intestinal cells) at 10 and 50 multiplicities of infection (MOI) influence RAW 264.7 macrophage activity, including nitric oxide (NO) production, migration, phagocytosis, bacterial killing, and the expression of cytokines (IL-6, IL-1β, TNF-α) and the chemokine CCL2. *C. jejuni* infection of HT-29 cells at 10 MOI induced significant IFN-γ production, a key macrophage activator. The treatment of macrophages with supernatants from HT-29 cells infected with *C. jejuni* significantly increased NO production, enhanced migration and phagocytic activity, and increased IL-6, TNF-α and CCL2 gene expression. However, no significant killing of phagocytosed *C. jejuni* was observed. On the other hand, supernatants from HT-29 cells infected with 50 MOI of *C. jejuni* suppressed NO production and macrophage phagocytosis, which may explain individual variations in the immune system’s ability to contain infection, potentially influenced by the infectious dose. These findings support the notion that *Campylobacter* can evade macrophage killing even under activated conditions. Further studies are needed to elucidate the molecular mechanisms by which *Campylobacter* survives within activated macrophages.

## 1. Introduction

*Campylobacter* a Gram-negative, motile bacterium, is a leading bacterial cause of gastroenteritis in humans worldwide [1]. It is primarily transmitted through the consumption of contaminated animal products containing a low infectious dose of approximately 500 colony-forming units, causing an estimated 1.5 million cases of campylobacteriosis annually in the US, with associated healthcare costs ranging from $2 to $11 billion [1,2,3,4]. Campylobacteriosis in healthy individuals is typically a self-limiting disease; however, *Campylobacter* infections in immunocompromised and elderly individuals can lead to severe clinical manifestations, such as reactive arthritis, inflammatory bowel disease, and Guillain-Barré syndrome, necessitating antimicrobial treatment [5]. To date, there is a limited understanding of host immune surveillance in response to this pathogen and its ability to cause disease, which has hindered the development of innovative strategies for its prevention and control in humans.

Upon entering the intestinal tract, *Campylobacter* expresses several virulence factors, including motility-associated genes (*FlaA*, *FlaB*), adhesion genes (*CadF*, *FlpA*), and invasion genes (*CiaB*, *iamA*), which facilitate motility, adhesion to and invasion of intestinal epithelial cells (IECs) [6]. Following adhesion to IECs, bacteria either engage their PAMPs—such as lipooligosaccharide, flagellin, and lipoproteins—with transmembrane pattern recognition receptors (PRRs), particularly Toll-like receptors (TLRs), or are internalized by IECs, where they release outer membrane vesicles (OMVs) [7]. These OMVs, together with bacterial PAMPs, engage cytosolic PRRs, including NOD-like receptors (NLRs) [7,8]. Both pathways trigger intracellular signaling cascades that promote the secretion of cytokines and chemokines, such as interleukin (IL)-1β, IL-6, IL-8, and tumor necrosis factor-alpha (TNF-α), thereby initiating inflammation and recruiting additional immune cells [7]. In addition, *Campylobacter* has been reported to induce IFN-γ secretion from IECs, which contributes to macrophage activation, leading to the production of nitric oxide (NO) and reactive nitrogen species (RNS), both of which are known to possess potent bactericidal activity [7,9].

During infection, *Campylobacter* secretes cytolethal distending toxin [10], which also triggers inflammatory cytokine production, drives intestinal pathology and contributes to the epithelial barrier disruption [11,12,13]. It then uses its corkscrew-like motility, mediated by polar flagella, to traverse the damaged epithelium and reach the lamina propria, where other innate immune cells reside.

While neutrophils contribute to the clearance of bacterial pathogens, macrophages are believed to play a particularly critical role in *Campylobacter* infection because of their ability to efficiently phagocytose whole bacterial cells that cross the intestinal epithelial barrier and coordinate adaptive immunity [7,14]. Previous experimental studies by our group and others indicate that in vitro-cultured macrophages can phagocytose *Campylobacter*; however, whether *Campylobacter* can survive within macrophages or is rapidly killed after internalization remains controversial [6,15,16,17]. For instance, while Banfi and colleagues suggested that macrophages can eliminate *Campylobacter* through both intracellular and extracellular killing mechanisms [15], other studies have reported that *Campylobacter* can survive and even replicate within human macrophages after phagocytosis [15,17]. This discrepancy may be due to strain-specific or host-specific differences or differences in macrophage activation status. Nonetheless, these studies typically assess the phagocytic activity of cultured macrophages without activation by immunomodulators secreted by other host cells in vivo.

In addition to investigating *Campylobacter*–macrophage interactions, studies using IECs have delineated several virulence factors associated with *Campylobacter* motility, adhesion, and invasion, as well as the innate immune responses initiated upon bacterial entry into the intestinal tract. While investigations focused on a single innate immune cell type—such as IECs or macrophages—have provided important insights into *Campylobacter* virulence and host responses, they do not fully capture the complexity of cellular interactions during infection.

Therefore, this study was conducted to evaluate the in vitro interactions between IECs and macrophages during *Campylobacter* infection, providing a more comprehensive model of early *Campylobacter*–host interactions. Specifically, we examined how soluble factors secreted by *Campylobacter*-infected IECs influence macrophage NO production, migration, phagocytosis and killing of *Campylobacter*, as well as their cytokine and chemokine secretion patterns.

## 2. Materials and Methods

### 2.1. Preparation of Campylobacter Culture

A pure culture of *C. jejuni* strain 81-176 was prepared as described previously [18]. Briefly, Brain Heart Infusion (BHI) agar, containing Preston selective *Campylobacter* supplement (Oxoid, Basingstoke, Hampshire, UK), was streaked with a frozen loopful of *C. jejuni*. Following streaking, the plates were incubated in microaerophilic conditions (10% CO_2_, 5% O_2_, and 85% N_2_) at 37 °C for 48 h. A 5 mL freshly prepared BHI broth was then inoculated with a few numbers of colonies, followed by re-incubation under the same microaerobic conditions for 48 h at 37 °C. Afterwards, 0.5 mL of this culture was transferred to 50 mL of BHI broth, incubated under the same conditions for 48 h. Subsequently, the bacterial suspension underwent centrifugation at 3500× *g* for 10 min. The pellet was then resuspended in phosphate-buffered saline (PBS, pH 7.4). The optical density of the suspension was measured at 600 nm using a spectrophotometer (VWR, Radnor, PA, USA), and the corresponding colony-forming units (CFU)/mL were calculated using the growth formula.

### 2.2. Stimulation of HT-29 Cells with Campylobacter

HT-29 cells (human colorectal adenocarcinoma cells), provided by Dr. Jeremy Tzeng, Department of Biological Sciences at Clemson University, were seeded in six replicates in a 24-well plate (Corning Inc., Corning, NY, USA) at a density of 1 × 10^6^ cells/well in 1 mL of DMEM containing 4.5 g/L glucose, L-glutamine, and sodium pyruvate, supplemented with 10% fetal bovine serum (FBS), 200 U/mL penicillin, and 80 µg/mL streptomycin and incubated at 37 °C in a humidified atmosphere with 5% CO_2_. The cells were allowed to adhere for 4 h, during which more than 90% became attached. Subsequently, the cells were infected with *C. jejuni* at a multiplicity of infection (MOI) of 10 or 50 in 1 mL of fresh DMEM per well or maintained in DMEM without *C. jejuni* as a negative control. *C. jejuni* was added to wells containing DMEM alone, without cells, at an MOI of 10 or 50. Cells were then incubated for 2, 6, 12, and 24 h at 37 °C in a humidified 5% CO_2_ incubator. However, prolonged incubation for 24 h resulted in excessive bacterial overgrowth, leading to complete destruction of the HT-29 cell monolayers. The culture media also turned yellow, indicating a drop in pH due to bacterial metabolism, which further compromised cell viability. Based on these observations, incubation times were limited to a maximum of 12 h. Following incubation, the plates were centrifuged at 300 × *g* for 3 min, and the supernatants were subsequently collected and filtered through a 0.45 μm syringe filter to remove bacterial cells. The supernatants from *Campylobacter* cultured in DMEM without cells were tested for their ability to induce NO production in RAW cells, in order to determine whether the induction was due to soluble factors secreted by *Campylobacter*-infected HT-29 cells or by *C. jejuni* itself.

### 2.3. Evaluation of the Cytotoxic Effects of Campylobacter on HT-29 Cells

The Colorimetric CyQUANT™ LDH Cytotoxicity Assay Kit (Thermo Fisher Scientific, Greenville County, SC, USA) was used to evaluate the effects of two concentrations of *C. jejuni* on the viability of HT-29 cells. Briefly, cells were seeded as described above and incubated with live *Campylobacter* at an MOI of 10 or 50. For spontaneous LDH release, triplicate wells were incubated with 10 µL of sterile medium. An additional five replicate wells of non-infected cells served as the negative control. After 2, 6, and 12 h of incubation at 37 °C, 10 µL of 10× lysis buffer was added to each well to determine maximum LDH release. LDH supplied in the kit was used as a positive control, in accordance with the manufacturer’s instructions. The plate was then incubated for 45 min at 37 °C, and the supernatants were collected. Cellular cytotoxicity was quantified in five independent HT-29 culture replicates, as per the manufacturer’s instructions. Optical density (OD) was measured at 490 nm, and the percentage of cytotoxicity was calculated using the following formula:% Cytotoxicity = ((Compound-treated LDH activity − Spontaneous LDH activity)/(Maximum LDH activity − Spontaneous LDH activity)) × 100

-Compound-treated LDH: LDH activity in the HT-29 supernatant-treated wells;-Spontaneous LDH: LDH activity in the distilled water-treated wells (baseline release);-Maximum LDH: LDH activity after complete lysis of cells (total release).

### 2.4. Evaluation of IFN-γ Production in Campylobacter-Infected HT-29 Cells

IFN-γ production by HT-29 cells in response to infection with 10 or 50 MOI of *C. jejuni* was measured in the supernatant of six replicates of infected and non-infected cells using the BD OptEIA™ Human IFN-γ ELISA Kit (BD Biosciences, San Jose, CA, USA), according to the manufacturer’s instructions.

### 2.5. Evaluation of NO Production in Macrophages Following Treatment with Supernatant from Campylobacter-Infected HT-29 Cells

NO production in RAW264.7 cells (monocyte/macrophage-like cells), provided by Dr. Alexis Stamatikos, Department of Food, Nutrition, and Packaging Sciences at Clemson University, was quantified following treatment with the supernatant of *C. jejuni*-infected HT-29 cells, using the Griess assay (Promega, Madison, WI, USA) following the manufacturer’s instructions. Briefly, RAW264.7 cells were seeded in five replicates at a density of 4 × 10^5^ cells/well in 500 μL of supplemented DMEM in 48-well plates and incubated for 3 h at 37 °C in a humidified 5% CO_2_ incubator to allow adherence, which typically exceeds 90%. After 3 h, the DMEM was replaced with supernatants from *Campylobacter*-infected HT-29 cells (containing host cell factors such as cytokines and chemokines and possible *C. jejuni* components such as secreted proteins, toxins, and OMVs) or cell-free *Campylobacter* supernatants (containing *C. jejuni* components), and cells were incubated for 2, 6, or 12 h under the same conditions. The supernatants were then collected for measuring NO.

To quantify NO, 50 μL of modified Griess reagent was added to 50 μL of the collected cell-free culture supernatants (from treated RAW cells or cell-free *Campylobacter* supernatants in DMEM) in a 96-well flat-bottom plate, yielding a final volume of 100 μL per well. After a 15-min incubation at room temperature, absorbance was measured at 540 nm using the PerkinElmer multimode plate reader. The concentration of nitrite (NO_2_^−^) released was calculated by interpolation from a standard curve of sodium nitrite (NaNO_2_).

### 2.6. Evaluation of Macrophage Migration Following Treatment with Supernatant from Campylobacter-Infected HT-29 Cells

The Transwell migration assay was performed as previously described [19] with slight modifications. Briefly, RAW264.7 cells were seeded in six replicates at a density of 1 × 10^5^ cells/100 μL DMEM containing 0.1% BSA, 4.5 g/L glucose, L-glutamine, and sodium pyruvate, supplemented with 10% fetal bovine serum (FBS), 200 U/mL penicillin, and 80 μg/mL streptomycin. Cells were plated on the membrane of a Transwell insert (Costar, Arlington, VA, USA) and incubated for 10 min at 37 °C in a humidified incubator with 5% CO_2_. To the lower chamber, 600 μL of supernatant derived from *C. jejuni*-infected HT29 cells (10 or 50 MOI) or control DMEM was added. After incubation for 24 h at 37 °C with 5% CO_2_, the media and non-migrated cells on the upper surface of the membrane were gently removed with a cotton swab. For fixation, the membrane was immersed in 800 μL of 70% ethanol for 10 min, followed by staining with 0.2% crystal violet for an additional 10 min. Excess dye was washed off with distilled water, and membranes were imaged using an inverted microscope. Migrated cells were quantified in four images captured from different non-overlapping fields using ImageJ software (version 1.54p).

### 2.7. Evaluation of Macrophage Phagocytic Activity Following Treatment with Supernatant from Campylobacter-Infected HT-29 Cells

The phagocytic activity of RAW264.7 cells was quantitatively assessed using a phagocytosis assay kit (Latex Beads Rabbit IgG-FITC; Cayman Chemical, Ann Arbor, MI, USA) following treatment with the supernatant from *C. jejuni*-infected HT-29 cells, as previously described [6]. Briefly, RAW264.7 cells were seeded in 96-well flat-bottom plates (4 × 10^5^ cells/well) in supplemented DMEM medium, with six replicates per treatment, and incubated for 3 h at 37 °C in a 5% CO_2_ incubator. Cells were then treated with supernatants derived from HT-29 cells infected with an MOI of 10 or 50, or with supernatants from non-infected control cells, in the presence of the Latex Beads Rabbit IgG-FITC complex. Following a 6-h incubation at 37 °C in a humidified 5% CO_2_ environment, the cells were centrifuged at 400× *g* for 10 min, and the supernatant was discarded. Fluorescence intensity was measured using a multimode plate reader (PerkinElmer, Shelton, CT, USA) at an excitation wavelength of 485 nm and an emission wavelength of 535 nm.

### 2.8. Evaluation of Macrophage Bactericidal Activity Against Campylobacter Following Treatment with Supernatant from Campylobacter-Infected HT-29 Cells

The gentamycin protection assay was performed as previously described [20], with minor modifications. Briefly, RAW264.7 cells were seeded at a density of 1 × 10^6^ cells/well in 1 mL of supplemented DMEM in 24-well plates and incubated for 3 h at 37 °C in a humidified 5% CO_2_ incubator to allow adherence. Two identical plates were prepared for two timepoints, T_0_ and T_2_, each containing three experimental groups with five replicates per group. Before treatment, the medium was aspirated, and cells were washed twice with PBS. Supernatants derived from *Campylobacter*-infected HT-29 cells were prewarmed to 37 °C and added to wells (1 mL/well), and cells were then infected with 10 MOI of *C. jejuni*. Plates were centrifuged at 286× *g* for 3 min to facilitate contact between *C. jejuni* and RAW264.7 cells, followed by incubation at 37 °C with 5% CO_2_. After 2 h, the medium was aspirated, and the cells were washed three times with PBS. Then, 1 mL DMEM containing 200 μg/mL gentamicin was added to each well and incubated for either 15 min (T_0_) or 2 h (T_2_). Following incubation, cells were washed three times with PBS to remove residual antibiotic, then lysed with 1 mL of filter-sterilized 1% saponin in distilled water per well for 10 min, followed by vigorous pipetting. Lysates from each well were serially diluted (10-fold, up to four dilutions) in PBS and plated on BHI agar supplemented with Preston *Campylobacter* Selective Supplement. Plates were incubated under microaerobic conditions (5% O_2_, 10% CO_2_, 85% N_2_) at 37 °C for 48 h. CFUs of *C. jejuni* were enumerated, and bacterial killing was calculated using the formula: Killing percentage = [(T0 − T2)/T0] × 100.

### 2.9. Evaluation of Gene Expressions of Cytokine and Chemokine in Macrophages Following Treatment with Supernatant from Campylobacter-Infected HT-29 Cells

#### 2.9.1. RNA Isolation and cDNA Preparation from Macrophages

Total RNA was extracted from six replicate macrophage cultures treated with supernatants from *C. jejuni*–infected HT-29 cells (MOI 10 or 50) for 2, 6, or 12 h using the Direct-zol RNA Kit (Zymo Research, Irvine, CA, USA), according to the manufacturer’s instructions. RNA concentration and purity were assessed using a Nanodrop One spectrophotometer (Thermo Fisher Scientific, Waltham, MA, USA). First-strand cDNA was synthesized from the isolated RNA using the Superscript^®^ II First-Strand Synthesis Kit (Invitrogen, Thermo Fisher Scientific, Waltham, MA, USA) with oligo(dT) primers (Thermo Fisher Scientific, Waltham, MA, USA) as previously described [21]. The resulting cDNA was diluted 1:10 in nuclease-free water (Thermo Fisher Scientific, Waltham, MA, USA) and stored at −20 °C until further use.

#### 2.9.2. Quantitative Real-Time PCR (qRT-PCR)

PCR amplification was carried out on a qTOWER^3^ real-time PCR system (Analytik Jena, Jena, Germany) using SYBR Green Master Mix (Thermo Fisher Scientific, Waltham, MA, USA) as previously described [22]. Each 20 µL reaction contained 10 µL of 2× SYBR Green mix, 1 µL each of forward and reverse primers diluted 1:10 from stock solutions of 100 µM, 5 µL of the template sample, and 3 µL of nuclease-free water. Primer sequences are provided in Table 1. Cycling conditions consisted of an initial denaturation at 95 °C for 2–3 min, followed by 40 cycles of 95 °C for 10–15 s, annealing at 55 °C for 20–30 s, and extension at 72 °C for 15–30 s. A final extension step was performed at 72 °C for 2–5 min. The specificity of amplification was confirmed by melt-curve analysis, performed from 65 to 95 °C with increments of 0.3–0.5 °C and 5–10 s holds per step. Relative gene expression was analyzed using the 2^−ΔΔCt^ method, with GAPDH serving as the reference control. Data were expressed as fold changes relative to the control group.

### 2.10. Statistical Analysis

Graphs were created using GraphPad Prism V5.0 (GraphPad Software, San Diego, CA, USA). Data from 5–6 biological replicates per assay were analyzed using one-way analysis of variance (ANOVA) to determine whether statistically significant differences existed among the treatment groups. When ANOVA indicated a significant effect, Tukey’s multiple comparison test was applied as a post hoc analysis to identify which specific groups differed from each other. Results were considered statistically significant if the *p* value ≤ 0.05. Data are graphically represented as the mean ± the standard error of the mean (SEM).

## 3. Results

### 3.1. Campylobacter-IECs Interactions

#### 3.1.1. Cytotoxic Effect of *Campylobacter* on IECs

This assay was performed to assess whether *C. jejuni* at two different concentrations (10 and 50 MOI) exerts cytotoxic effects on HT-29 cells using a lactate dehydrogenase (LDH) release assay. The results showed no significant differences in cytotoxicity between infected and non-infected cells, with overall values not exceeding 5% in the infected groups (Figure 1). Based on these findings, these two concentrations were used for subsequent experiments.

#### 3.1.2. *Campylobacter* Enhanced IFN-γ Production by IECs

The protein levels of IFN-γ were measured in the supernatant of *C. jejuni*-infected IECs as an indicator of immunomodulator secretion in response to *Campylobacter* and as a potent activator of macrophages. No significant differences were observed in IFN-γ levels at 2- and 6-h post-infection with 10 or 50 MOI of *C. jejuni*. However, a significant increase was detected at 12 h post-infection with both concentrations (Figure 2). Based on these observations, supernatants collected at the 12-h time point were selected for all subsequent macrophage assays, as this was the peak of IFN-γ production—a key macrophage activator—and it preceded the significant cell monolayer destruction and pH change observed at 24 h.

### 3.2. Campylobacter-Macrophage Interactions

#### 3.2.1. Soluble Factors Secreted by *Campylobacter*-Infected IECs Stimulate NO Production in Macrophages

As an indicator of macrophage activity, NO levels were measured in the macrophages at 2-, 6-, and 12-h following treatment with supernatants from *Campylobacter*-infected IECs or from *Campylobacter* incubated in the same culture medium but without cells. While the *Campylobacter* cell-free supernatant from medium without HT-29 cells—likely containing secreted proteins, peptides, metabolic by-products, OMVs, and lipooligosaccharides (LOS)—significantly reduced NO production at both concentrations, a significant increase in NO production was observed in response to the supernatant from *C. jejuni-infected* HT-29 cells at 10 MOI at both 2- and 12-h post-treatment. Treatment with the supernatant from 50 MOI *C. jejuni*-infected IECs significantly reduced NO production at both 6- and 12-h post-treatment, whereas no significant change was observed at 2 h (Figure 3).

#### 3.2.2. Soluble Factors Secreted by *Campylobacter*-Infected IECs Promote Macrophage Migration Activity

Treatment of macrophages with supernatants from *Campylobacter*-infected HT-29 cells at both 10 and 50 MOI significantly increased macrophage migration in a dose-dependent manner compared to macrophages treated with supernatants from non-infected cells. However, no significant difference was observed in macrophage migration between the 10 and 50 MOI (Figure 4).

#### 3.2.3. Soluble Factors Secreted by *Campylobacter*-Infected IECs Enhance Macrophage Phagocytic Activity

Treatment of macrophages with supernatants from *Campylobacter*-infected IECs at 10 MOI significantly enhanced phagocytosis at 6 h post-treatment, with no significant changes observed at 2 or 12 h, while treatment with the supernatant from IECs infected at 50 MOI significantly reduced macrophage phagocytic activity at 12 h post-treatment, emphasizing the dose-dependent effect (Figure 5).

#### 3.2.4. Soluble Factors Secreted by *Campylobacter*-Infected IECs Did Not Influence the Macrophage Killing Activity

To assess whether macrophage activation enhances their ability to kill *Campylobacter*, a gentamicin protection assay was performed. Although the killing percentage was numerically higher in macrophages treated with supernatants from *Campylobacter*-infected HT-29 cells at both 10 and 50 MOI (25% and 8%, respectively), it was not significantly different from that observed in macrophages treated with supernatants from non-infected cells (Figure 6). Raw data are provided in Appendix A.

#### 3.2.5. Soluble Factors Secreted by *Campylobacter*-Infected IECs Alter Cytokine and Chemokine Gene Expression in Macrophages

To gain insights into the immunomodulatory activity of soluble factors secreted by IECs in response to *Campylobacter*, we evaluated their effects on macrophage expression of proinflammatory cytokines (IL-1β, IL-6, and TNF-α) and the chemokine C-C motif chemokine ligand 2 (CCL2).

The treatment of macrophages with supernatants from *Campylobacter*-infected HT-29 cells at 10 or 50 MOI significantly increased IL-6 gene expression at 2 and 12 h, respectively, compared to macrophages treated with supernatants from non-infected cells. However, no significant differences were observed in the expression of this gene at 6 h post-treatment (Figure 7a).

With respect to IL-1β, a significant decrease in its expression was observed at 6 h post-treatment in the cells treated with supernatants from *Campylobacter*-infected HT-29 cells at 10 MOI, whereas a significant increase in expression was observed at 12 h following treatment with supernatants from *Campylobacter*-infected HT-29 cells at 50 MOI. No significant differences were observed in IL-1β expression at 2 h post-treatment (Figure 7b).

Similar patterns of significantly induced TNF-α (Figure 7c) and CCL2 (Figure 7d) gene expression were observed at 2-, 6-, and 12-h in the cells treated with supernatants from *Campylobacter*-infected HT-29 cells at 50 MOI. Treatment of macrophages with supernatants from *Campylobacter*-infected HT-29 cells at 10 MOI induced TNF-α expression only at 12 h post-treatment.

## 4. Discussion

Extensive in vitro research has examined interactions between *Campylobacter* and host immune cells, including IECs and macrophages [25,26,27,28]. Although these in vitro studies provide valuable insights into host–microbe interactions, including the identification of some mechanistic pathways and characterization of immune responses in these cells during *Campylobacter* infection, they do not fully recapitulate the in vivo gut environment. For example, most experiments have directly exposed in vitro-cultured macrophages to *Campylobacter* [25,26,27,28,29], overlooking potential pre- or co-activation by immunomodulatory molecules secreted by other immune cells, such as epithelial cells that first encounter the bacteria in the gut. Evaluating the interplay between *Campylobacter* and IECs, including the effects of their secreted molecules on macrophage activation, may offer deeper insight into immune activation than analyses focused on individual immune cell types. Therefore, this study was undertaken to evaluate the crosstalk between IECs and macrophages in an in vitro model, specifically investigating how IEC-secreted molecules in response to *Campylobacter* influence macrophage NO production, migration, phagocytosis, bacterial killing, and cytokine and chemokine secretion. IECs can release numerous immunomodulatory molecules in response to *Campylobacter*, including IFN-γ, TNF-α, and granulocyte-macrophage colony-stimulating factors (GM-CSF), all of which are known to stimulate macrophage activation [26,30]. IFN-γ was measured in particular due to its central role in promoting macrophage activation, M1 polarization, and antigen presentation [31,32].

To rule out cytotoxic effects of *Campylobacter* toxins on IECs, an LDH release assay was performed, revealing no significant differences between infected and non-infected cells. Exposure of IECs to *Campylobacter* significantly induced IFN-γ production, prompting an assessment of whether supernatants from these cells could enhance macrophage activity. Notably, supernatants from *Campylobacter*-infected IECs induced significant NO production in macrophages, whereas supernatants from *Campylobacter* incubated in medium without IECs suppressed NO production at both concentrations. This underscores the importance of IEC-macrophage crosstalk: while direct exposure of macrophages to *Campylobacter*-secreted proteins, such as proteins, toxins, and OMVs, suppresses NO, which could be a potential immune evasion strategy, the secreted molecules from IECs, such as cytokines, chemokines and other immunomodulatory molecules, counteract this effect. This could explain the eventual control of the infection in self-limiting cases [7,26,33]. Moreover, the suppression of NO production and macrophage phagocytosis in cells treated with supernatants from 50 MOI-infected IECs is likely due to the presence of higher levels of specific bacterial immunosuppressive factors in the cell-free medium, which may account for dose-dependent variations in the immune system’s ability to control infection [34]. However, since the in vitro model does not fully capture the complexities of the host response, further validation of these findings in a dose–response study in an animal infection model is warranted.

In the present study, macrophages migrated in a dose-dependent manner toward supernatants from *Campylobacter*-infected IECs, suggesting that infected IECs release chemoattractants that initiate macrophage recruitment before bacterial recognition and phagocytosis. This observation is consistent with earlier studies showing that *Campylobacter*-derived PAMPs stimulate human IECs to secrete IL-8 [11,13], which in turn recruits innate immune cells, including neutrophils and macrophages, to the site of infection [7,35,36].

Accumulating evidence highlights the role of macrophage phagocytosis in host defense against *Campylobacter* [35,37]. Yet, whether phagocytosed bacteria can resist killing and survive inside macrophages remains controversial, possibly because many in vitro assays lack co-activation of macrophages by secreted immunomodulatory products from other immune cells. In this study, co-activation of macrophages with secreted molecules from *Campylobacter*-infected IECs did not enhance bacterial killing despite increased NO production and phagocytosis. However, this finding should be interpreted with caution, as the lack of a statistically significant enhancement in bacterial killing may reflect limited assay sensitivity or a minimal biological effect. Another important consideration is the relatively short incubation periods in our study, where IECs were exposed to *Campylobacter* for a maximum of 12 h, which may not have been sufficient for the cells to fully release their modulatory molecules. Similarly, short-term exposure of HT-29 cells to supernatants from infected cells may not accurately capture the longer-term interactions that typically occur over a few days in an infected host. These observations are consistent with those of Anna Duda-Madej et al. (2025), who reported that a 24-h exposure to berberine was insufficient to confer a protective effect on colonocytes exposed to *C. jejuni* post-culture supernatant, with statistically significant differences observed only after 72 h of incubation [4]. Nonetheless, these findings provide insight into the pathogen’s ability to cause gastroenteritis in healthy individuals and support the notion that *Campylobacter* can persist intracellularly in macrophages for extended periods by subverting phagolysosomal killing [38].

The secretory molecules from *C. jejuni*-infected HT-29 cells at both 10 and 50 MOI induced macrophage expression of pro-inflammatory cytokines, including IL-1β, IL-6, and TNF-α, as well as the chemokine CCL2, in a dose- and time-dependent manner, all of which have been previously reported to contribute to the initiation of inflammation and the development of gastroenteritis [7]. Nevertheless, despite a robust pro-inflammatory gene expression profile (IL-6, TNF-α and CCL2) induced by epithelial signals, enhanced phagocytosis, and increased NO production, this activation was insufficient to enhance bacterial killing, highlighting the need to investigate additional molecular and cellular mechanisms underlying *Campylobacter* survival within activated macrophages.

Although this study did not investigate the underlying molecular mechanisms of IECs and macrophage activation, it provides valuable insight into the importance of IEC–macrophage interactions in controlling *Campylobacter* infection. It also emphasizes the significance of cellular communication in in vitro assays for understanding host–pathogen interactions, rather than studying individual cell types separately. Additionally, it suggests how different infectious doses of *Campylobacter* may influence the immune system’s capacity to respond to and clear infection. Future studies will include mass spectrometry-based profiling of the secreted molecules in the supernatants from *Campylobacter*-infected HT-29 cells, along with targeted cytokine-neutralization assays in macrophages, to identify the specific mediators responsible for macrophage activation. Additionally, incorporating protein-level analyses, such as ELISA, will enable the quantification of secreted cytokines and confirmation of the magnitude of the observed immune responses.

## 5. Conclusions

This study highlights the interplay between IECs and macrophages during *Campylobacter* infection, showing that macrophages can mount diverse immune responses to IEC-derived signals while still being unable to fully kill phagocytosed bacteria. This may explain why *Campylobacter* proliferates in the gut and causes inflammation despite an active immune response. Further research is needed to elucidate the mechanisms behind these interactions. For example, identifying the survival mechanisms of *Campylobacter* within macrophages, as well as the signaling molecules involved in macrophage activation and phagocytosis, could inform the development of future immune-based therapeutic interventions.

## Figures and Tables

**Figure 1 microorganisms-13-02808-f001:**
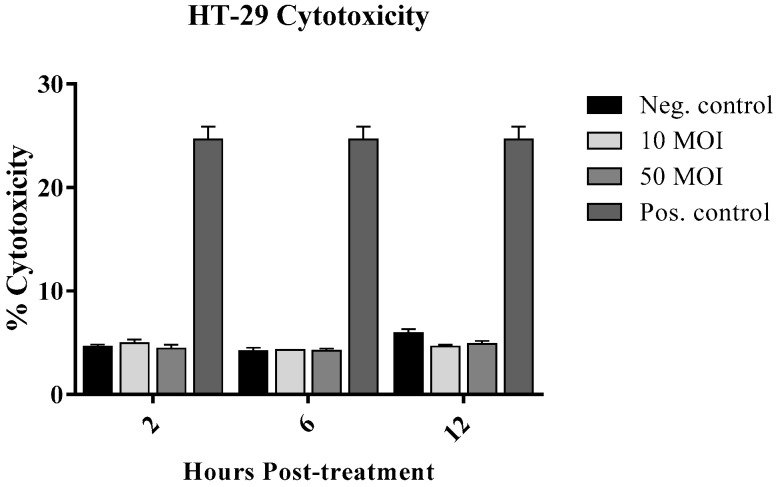
Cytotoxicity of HT-29 cells was measured by the colorimetric CyQUANT™ LDH cytotoxicity assay kit in five biological replicates of HT-29 cells following infection with *Campylobacter jejuni* at 10 or 50 MOI for 2, 6, and 12 h. Data were analyzed using one-way analysis of variance (ANOVA), followed by Tukey’s multiple comparison test. No significant differences were observed between the infected and non-infected cells. This assay was repeated twice, yielding similar results with no significant differences.

**Figure 2 microorganisms-13-02808-f002:**
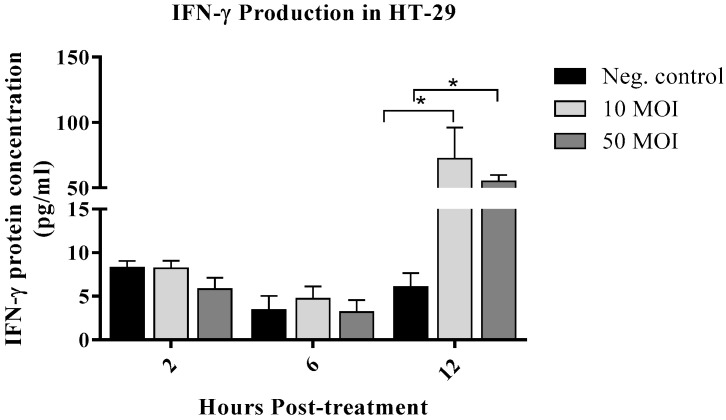
Interferon (IFN)-γ levels in six biological replicates of HT-29 cells following infection with *Campylobacter jejuni* at 10 or 50 MOI for 2, 6, and 12 h. Data were analyzed using one-way analysis of variance (ANOVA), followed by Tukey’s multiple comparison test. Results are presented as the mean ± SEM. Asterisks (*) indicate statistically significant increases in IFN-γ production (*p* < 0.05).

**Figure 3 microorganisms-13-02808-f003:**
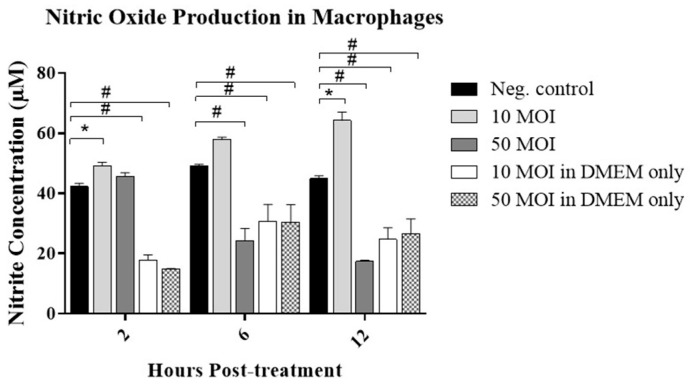
Nitric oxide (NO) production by RAW 264.7 cells following treatment with the supernatant from five biological replicates of *C. jejuni*–infected HT-29 cells at 10 or 50 MOI or *Campylobacter* cell-free supernatant for 2-, 6-, and 12-h. Data were analyzed using one-way analysis of variance (ANOVA), followed by Tukey’s multiple comparison test. Results are presented as the mean ± SEM. Asterisks (*) indicate statistically significant increases in NO production (*p* < 0.05), while pound symbols (#) indicate statistically significant decreases (*p* < 0.05).

**Figure 4 microorganisms-13-02808-f004:**
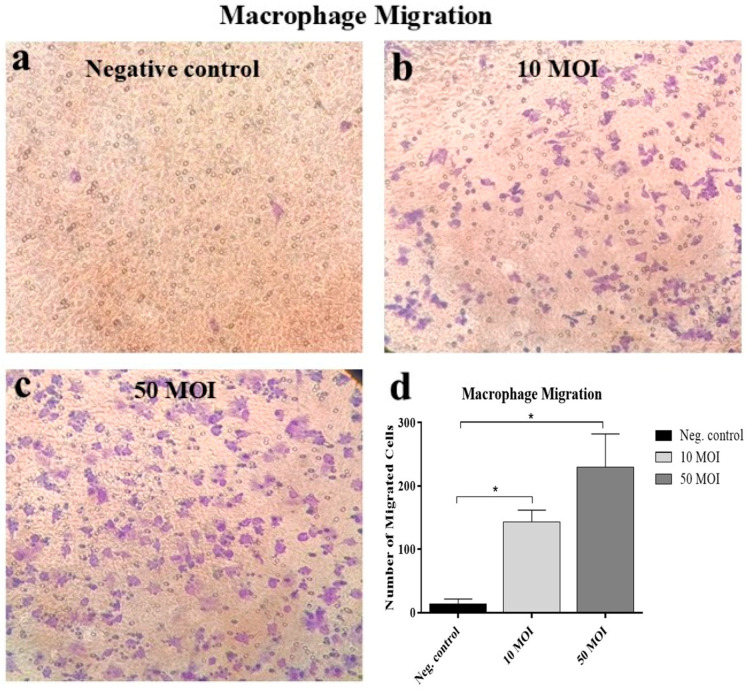
RAW 264.7 cell migration across the transmembrane in response to supernatants from six biological replicates of HT-29 cultures infected with *Campylobacter* (MOI 10 or 50) for 24 h. Migrated cells were stained with crystal violet in the negative control/untreated group (**a**), cells treated with supernatants from HT-29 cells infected with *Campylobacter* at MOI 10 (**b**), and cells treated with supernatants from HT-29 cells infected at MOI 50 (**c**). Panel (**d**) shows quantification of migrated cells across multiple fields. Data were analyzed using one-way analysis of variance (ANOVA), followed by Tukey’s multiple comparison test. Results are presented as the mean ± SEM. Asterisks (*) indicate statistically significant increases in cell migration (*p* < 0.05). The experiment was repeated twice, yielding a similar migration pattern.

**Figure 5 microorganisms-13-02808-f005:**
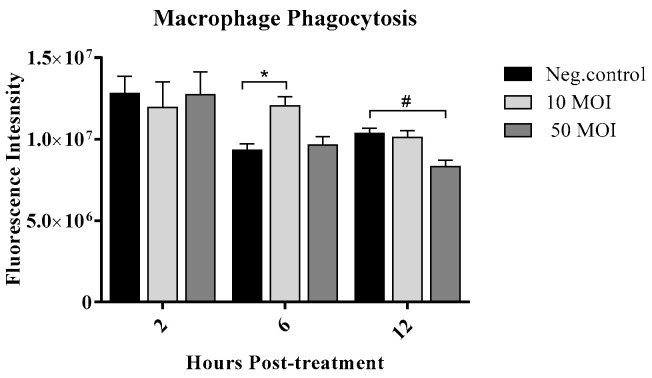
Phagocytic activity of RAW 264.7 cells was measured using the Latex Beads Rabbit IgG-FITC kit following treatment with supernatants from six biological replicates of HT-29 cultures infected with *Campylobacter jejuni* at MOI 10 or 50 for 2-, 6-, and 12-h. Fluorescence intensity was measured using a plate reader with an excitation wavelength of 485 nm and an emission wavelength of 535 nm. The higher fluorescence indicates higher phagocytosis. Data were analyzed using one-way analysis of variance (ANOVA), followed by Tukey’s multiple comparison test. Results are presented as the mean ± SEM. Asterisks (*) indicate statistically significant increases in macrophage phagocytic activity (*p* < 0.05), while the pound symbol (#) indicates statistically significant decreases (*p* < 0.05).

**Figure 6 microorganisms-13-02808-f006:**
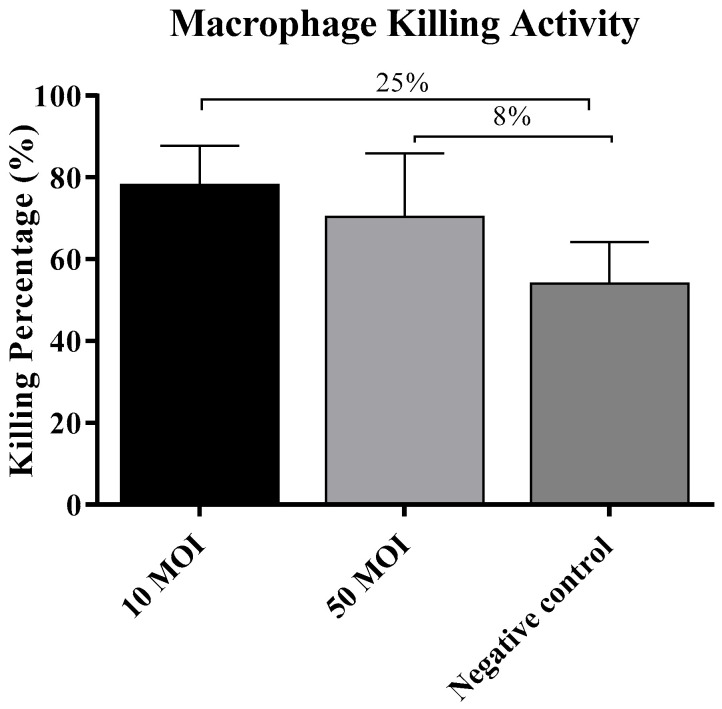
Macrophage killing activity against *Campylobacter jejuni* was assessed using the gentamicin protection assay after treatment with supernatants from five biological replicates of HT-29 cultures infected with *C. jejuni* at MOI 10 or 50 for 12-h. Data were analyzed using one-way analysis of variance (ANOVA), followed by Tukey’s multiple comparison test. Results are presented as the mean ± SEM. The experiment was repeated twice with consistent, non-significant results.

**Figure 7 microorganisms-13-02808-f007:**
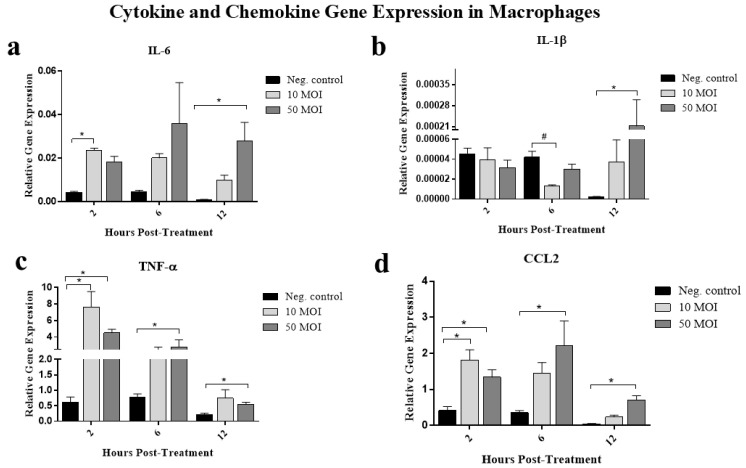
Gene expression of interleukin (IL)-6 (**a**), IL-1β (**b**), tumor necrosis factor alpha (TNF-α) (**c**) and C-C motif chemokine ligand 2 (CCL2) (**d**) was measured in six biological replicates of RAW 264.7 cells using quantitative real-time PCR (qRT-PCR) following treatment with supernatants from *C. jejuni*–infected HT-29 cells (MOI 10 or 50) for 2, 6, or 12 h. Data were analyzed using one-way analysis of variance (ANOVA), followed by Tukey’s multiple comparison test. Results are presented as mean ± SEM. Asterisks (*) indicate statistically significant increases in gene expression (*p* < 0.05), while pound symbols (#) indicate statistically significant decreases (*p* < 0.05).

**Table 1 microorganisms-13-02808-t001:** Primer sequences used for quantitative real-time qRT-PCR.

Gene	Forward Primer (5′→3′)	Reverse Primer (5′→3′)	Reference
*GAPDH*	*TGACCTCAACTACATGGTCTACA*	*CTTCCCATTCTCGGCCTTG*	[23]
*CCL2*	*GTCCCTGTCATGCTTCTG*	*CTGCTGGTGATCCTCTTG*	[23]
*IL-6*	*TCTATACCACTTCACAAGTCGGA*	*GAATTGCCATTGCACAACTCTTT*	[24]
*IL-1β*	*GAAATGCCACCTTTTGACAGTG*	*TGGATGCTCTCATCAGGACAG*	[24]
*TNF-α*	*CAGGCGGTGCCTATGTCTC*	*CGATCACCCCGAAGTTCAGTAG*	[24]

## Data Availability

The original contributions presented in this study are included in the article/Appendix A. Further inquiries can be directed to the corresponding author.

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
