# Peer review of "Epithelial–Macrophage Crosstalk in Host Responses to Campylobacter jejuni Infection in Humans"

_microorganisms, 2025, doi:10.3390/microorganisms13122808_

Round 1
Reviewer 1 Report (Previous Reviewer 4)
Comments and Suggestions for Authors
This manuscript, titled "Epithelial–Macrophage Crosstalk in Host Responses to Cam- pylobacter Infection in Humans," primarily investigates the interactions between intestinal epithelial cells (IECs) and macrophages during Campylobacter jejuni infection. Through in vitro models such as HT-29 intestinal epithelial cells and RAW 264.7 macrophages, it was found that soluble factors secreted by epithelial cells after infection, such as IFN - γ, can significantly affect macrophage activity, including enhancing nitric oxide (NO) production, migration ability, and phagocytosis. However, high infectious doses (50 MOI) can inhibit certain functions such as NO production and phagocytosis; Although macrophages were activated, their bacterial killing ability did not significantly increase, suggesting that Campylobacter may survive inside cells through immune escape mechanisms. The research significance lies in emphasizing the core role of epithelial macrophage crosstalk in host immune response, providing a more comprehensive perspective on infection dynamics than single cell models, revealing dose-dependent immune regulation differences, and providing an important theoretical basis for understanding the pathogenesis of Campylobacter disease and developing new therapies targeting immune escape. However, there are still areas in the manuscript that need improvement, such as:
- In the introduction, it is mentioned that Campylobacter infection can cause severe clinical symptoms in patients or elderly people with weakened immune function. However, please provide examples to supplement the specific manifestations;
- It is recommended to indicate the number of biological replicates in the legend of each image;
- All microscopic images should be added with a scale bar and the length of the scale bar should be indicated in the corresponding legend;
Although this study detected changes in macrophage function and cytokine gene expression, it did not clarify which intracellular signaling pathways were activated. It is recommended to supplement Western Blot experiments to analyze key signaling molecules;
- In the discussion section, it is pointed out that further research is needed to elucidate the mechanisms behind the interaction between macrophages and Campylobacter jejun. Please provide examples of how to conduct further research and what direction to start from?
Author Response
We sincerely appreciate the positive feedback from the reviewers and their valuable comments. Below, we provide a point-by-point response to each of their concerns.
- In the introduction, it is mentioned that Campylobacter infection can cause severe clinical symptoms in patients or elderly people with weakened immune function. However, please provide examples to supplement the specific manifestations;
As per the reviewer’s suggestion, we have included the following examples in the manuscript:
“Campylobacter infections in immunocompromised and elderly individuals can lead to severe clinical manifestations, such as reactive arthritis, inflammatory bowel disease, and Guillain-Barré syndrome, necessitating antimicrobial treatment.”
- It is recommended to indicate the number of biological replicates in the legend of each image;
Thank you for your suggestion. We have included the number of biological replicates in all figure legends.
- All microscopic images should be added with a scale bar and the length of the scale bar should be indicated in the corresponding legend;
Unfortunately, we did not have a microscope with this capability available in our laboratory. However, all images were captured using the same magnification settings. We would also like to highlight that a scale bar is not necessary for cell counts, as these analyses were based on consistent magnification rather than absolute dimensional measurements.
- Although this study detected changes in macrophage function and cytokine gene expression, it did not clarify which intracellular signaling pathways were activated. It is recommended to supplement Western Blot experiments to analyze key signaling molecules;
We agree with the reviewer that further investigation into the immunological mechanisms underlying these interactions, as well as the use of Western blotting to identify signaling molecules, would be valuable. However, we would like to emphasize that, although these experiments were not performed in the present study, our findings provide important insights into the general interactions between epithelial cells and macrophages during Campylobacter infection. These results lay the groundwork for future studies aimed at dissecting the underlying molecular mechanisms.
- In the discussion section, it is pointed out that further research is needed to elucidate the mechanisms behind the interaction between macrophages and Campylobacter jejun. Please provide examples of how to conduct further research and what direction to start from?
Thank you for your valuable input. We have included the following examples for future directions: “For example, identifying the survival mechanisms of Campylobacter within macrophages, as well as the signaling molecules involved in macrophage activation and phagocytosis, could inform the development of future immune-based therapeutic interventions.”
We appreciate any further input from the reviewer.
Reviewer 2 Report (Previous Reviewer 3)
Comments and Suggestions for Authors
This is a well-written and scientifically sound manuscript that addresses an important gap in our understanding of Campylobacter jejuni host-pathogen interactions. The study design is logical, the experiments are generally well-described, and the findings on the dose-dependent, dual role of epithelial-derived signals are novel and significant.
Here is a point-by-point revision to help strengthen the manuscript for publication.
MAJOR REVISION POINTS
1. Clarify the Experimental Setup and Supernatant Composition.
This is the most critical point for the reader's understanding. The manuscript needs to explicitly state what is contained in the key experimental reagent—the supernatant from infected HT-29 cells.
-
Suggestion: In the Methods (Section 2.2) and/or Results, explicitly clarify that the "supernatant from C. jejuni-infected HT-29 cells" is a mixture containing:
-
Host cell factors (cytokines, chemokines) secreted by HT-29 cells in response to infection.
-
Possible bacterial components (e.g., secreted proteins, toxins, OMVs) released by C. jejuni during the 12-hour co-culture with HT-29 cells.
-
Metabolic by-products from both cell types.
This clarification is vital for the correct interpretation of Figure 3. The suppression of NO by the "cell-free Campylobacter supernatant" (bacteria in DMEM alone) is a key control, but the supernatant from infected HT-29 is a more complex mixture. The discussion should reflect this complexity.
-
2. Justify the Choice of the 12-hour Time Point More Robustly.
The decision to use the 12-hour supernatant for all macrophage assays is based solely on IFN-γ data. However, other critical factors influencing macrophage behavior (other cytokines, bacterial load, host cell damage) also evolve over time.
-
Suggestion: In the Results (Section 3.1.2) or Methods, add a sentence justifying this choice more comprehensively. For example:
*"Supernatants collected at the 12-hour time point were selected for all subsequent macrophage assays, as this was the peak of IFN-γ production—a key macrophage activator—and it preceded the significant cell monolayer destruction and pH change observed at 24 hours."*
3. Reconcile the Gene Expression and Functional Data in the Discussion.
The discussion effectively covers the functional data (phagocytosis, NO, migration) but does not fully integrate the cytokine/chemokine gene expression results (Figure 7). There is a disconnect between the strong pro-inflammatory gene signature and the lack of bacterial killing.
-
Suggestion: Expand the discussion in the paragraph covering Figure 7. Explicitly state that despite a robust pro-inflammatory gene expression profile (IL-6, TNF-α, CCL2) induced by the epithelial signals, this activation was insufficient to enhance bacterial killing. This strengthens the central conclusion about C. jejuni's ability to evade intracellular killing even in an "activated" environment.
4. Improve the Statistical Reporting and Data Presentation.
-
Figure 6 (Killing Assay): The lack of statistical significance is mentioned, but the data presentation is weak. A bar graph showing non-significant differences with raw data points overlaid would be more informative than just stating percentages in the text.
-
Clarity on Replicates: The text sometimes mentions "five-six replicates" and "technical duplicates." It is best to consistently state the number of biological replicates (n) for each experiment, as this is crucial for assessing statistical power.
MINOR REVISION POINTS
1. Title and Abstract:
-
Title: Consider specifying the species (C. jejuni) for precision: "Epithelial–Macrophage Crosstalk in Host Responses to Campylobacter jejuni Infection."
-
Abstract, Line 22: "The treatment of macrophages with supernatants... significantly increased NO production, enhanced migration and phagocytic activity, and increased IL-6, TNF-α and CCL2 gene expression." (Add "increased" for parallel structure).
2. Introduction:
-
Line 59: "In addition, Campylobacter induces IFN-γ secretion from IECs..." This is a strong claim for the introduction, as your study demonstrates it. Consider softening to: "In addition, Campylobacter has been reported to induce IFN-γ secretion..." or "In our model, Campylobacter induced..."
-
Line 76-77: "This discrepancy may be due to strain-specific or host-specific differences." This is a good point. You could also add "or differences in macrophage activation status," which directly links to the rationale of your study.
3. Methods:
-
Section 2.2: "After 2, 6, 12 and 24 h of incubation..." -> "After 2, 6, 12, and 24 h of incubation..." (Add comma).
-
Section 2.8: There are two sections labeled "2.8". The second one (Gene Expression) should be "2.9". All subsequent numbering in the Methods should be updated accordingly.
-
Section 2.9 (Statistical Analysis): Specify that data are presented as mean ± SEM.
4. Results:
-
Figure 3 Legend: It says "five replicates," but the text says the assay was performed in "technical duplicates." Clarify the number of biological replicates.
-
Figure 5: The Y-axis label "Fluorescence Intensity" is clear, but a note explaining that higher fluorescence indicates higher phagocytosis would be helpful for a broader audience.
-
Section 3.2.5: When describing Figure 7, be precise. E.g., "a significant decrease was observed... at 6 hours post-treatment... at 10 MOI, whereas a significant increase... was observed at 12 hours... at 50 MOI."
5. Discussion:
-
Line 458-459: "...the immune system can often partially control infection." Consider rephrasing for accuracy, as the infection is often self-limiting, but the immune system doesn't necessarily "control" it before symptoms arise. Suggest: "...this may contribute to the eventual control of the infection in self-limiting cases."
-
Line 480-482: "However, this finding should be interpreted with caution, as the lack of a statistically significant enhancement in bacterial killing may reflect limited assay sensitivity or a minimal biological effect." This is good, objective writing. Keep it.
6. Formatting and Typos:
-
Line 2: "Cam- 2 pylobacter" -> "Campylobacter"
-
Line 10: "Animal and Veterinary Sci- 10 ences" -> "Sciences"
-
Line 27: "C. jejunishowed" -> "C. jejuni showed"
-
Keywords (Line 36): Capitalize nouns as per journal style (typically all nouns are capitalized): "Macrophage; Epithelial Cell; Intestine; Immunity; Campylobacter; Communication; Crosstalk."
Author Response
MAJOR REVISION POINTS
- Clarify the Experimental Setup and Supernatant Composition.
This is the most critical point for the reader's understanding. The manuscript needs to explicitly state what is contained in the key experimental reagent—the supernatant from infected HT-29 cells.
- Suggestion: In the Methods (Section 2.2) and/or Results, explicitly clarify that the "supernatant from C. jejuni-infected HT-29 cells" is a mixture containing:
- Host cell factors (cytokines, chemokines) secreted by HT-29 cells in response to infection.
- Possible bacterial components (e.g., secreted proteins, toxins, OMVs) released by C. jejuni during the 12-hour co-culture with HT-29 cells.
- Metabolic by-products from both cell types.
This clarification is vital for the correct interpretation of Figure 3. The suppression of NO by the "cell-free Campylobacter supernatant" (bacteria in DMEM alone) is a key control, but the supernatant from infected HT-29 is a more complex mixture. The discussion should reflect this complexity.
We thank the reviewer for their valuable input: we have modified the statement to the following:
“After 3 h, the DMEM medium was replaced with supernatants from Campylobacter-infected HT-29 cells (containing host cell factors such as cytokines and chemokines and possible C. jejuni components such as secreted proteins, toxins, and OMVs) or cell-free Campylobacter supernatants (containing C. jejuni components), and cells were incubated for 2, 6, or 12 h under the same conditions.”
We have also revised the corresponding statement in the discussion to the following “his underscores the importance of IEC-macrophage crosstalk: while direct exposure of macrophages to Campylobacter-secreted proteins, such as proteins, toxins, and OMVs, suppresses NO, which could be a potential immune evasion strategy, the secreted molecules from IECs, such as cytokines, chemokines and other immunomodulatory molecules, counteract this effect.”
- Justify the Choice of the 12-hour Time Point More Robustly.
The decision to use the 12-hour supernatant for all macrophage assays is based solely on IFN-γ data. However, other critical factors influencing macrophage behavior (other cytokines, bacterial load, host cell damage) also evolve over time.
- Suggestion: In the Results (Section 3.1.2) or Methods, add a sentence justifying this choice more comprehensively. For example:
*"Supernatants collected at the 12-hour time point were selected for all subsequent macrophage assays, as this was the peak of IFN-γ production—a key macrophage activator—and it preceded the significant cell monolayer destruction and pH change observed at 24 hours."*
Thank you for your valuable suggestion. We have revised the statement in the results section to the following: “Based on these observations, supernatants collected at the 12-hour time point were selected for all subsequent macrophage assays, as this was the peak of IFN-γ production—a key macrophage activator—and it preceded the significant cell monolayer destruction and pH change observed at 24 hours.”
- Reconcile the Gene Expression and Functional Data in the Discussion.
The discussion effectively covers the functional data (phagocytosis, NO, migration) but does not fully integrate the cytokine/chemokine gene expression results (Figure 7). There is a disconnect between the strong pro-inflammatory gene signature and the lack of bacterial killing.
- Suggestion: Expand the discussion in the paragraph covering Figure 7. Explicitly state that despite a robust pro-inflammatory gene expression profile (IL-6, TNF-α, CCL2) induced by the epithelial signals, this activation was insufficient to enhance bacterial killing. This strengthens the central conclusion about C. jejuni's ability to evade intracellular killing even in an "activated" environment.
Thank you. We have revised the statement to the following: “Despite a robust pro-inflammatory gene expression profile (IL-6, TNF-α, CCL2) induced by epithelial signals, enhanced phagocytosis, and increased NO production, this activation was insufficient to enhance bacterial killing, highlighting the need to investigate additional molecular and cellular mechanisms underlying Campylobacter survival within activated macrophages.”
- Improve the Statistical Reporting and Data Presentation.
- Figure 6 (Killing Assay): The lack of statistical significance is mentioned, but the data presentation is weak. A bar graph showing non-significant differences with raw data points overlaid would be more informative than just stating percentages in the text.
We appreciate the reviewer’s suggestion regarding data presentation. We would like to emphasize that we reported percentages in the main text to clearly convey the extent to which activated macrophages are capable of killing Campylobacter, providing a straightforward and interpretable measure for readers. To further enhance transparency and allow in-depth assessment, we have included a supplementary table presenting the raw colony count data. This approach balances clarity in the main text with access to detailed quantitative information, ensuring both readability and scientific rigor.
- Clarity on Replicates: The text sometimes mentions "five-six replicates" and "technical duplicates." It is best to consistently state the number of biological replicates (n) for each experiment, as this is crucial for assessing statistical power.
Thank you for your suggestion. We have included the number of biological replicates in all figure legends.
MINOR REVISION POINTS
- Title and Abstract:
- Title: Consider specifying the species (C. jejuni) for precision: "Epithelial–Macrophage Crosstalk in Host Responses to Campylobacter jejuni Infection."
We have included “jejuni” in the title.
- Abstract, Line 22: "The treatment of macrophages with supernatants... significantly increased NO production, enhanced migration and phagocytic activity, and increased IL-6, TNF-α and CCL2 gene expression." (Add "increased" for parallel structure).
We add the word increased as suggested by the reviewer.
- Introduction:
- Line 59: "In addition, Campylobacter induces IFN-γ secretion from IECs..." This is a strong claim for the introduction, as your study demonstrates it. Consider softening to: "In addition, Campylobacter has been reported to induce IFN-γ secretion..." or "In our model, Campylobacter induced..."
Thank you. We have made the suggested change
- Line 76-77: "This discrepancy may be due to strain-specific or host-specific differences." This is a good point. You could also add "or differences in macrophage activation status," which directly links to the rationale of your study.
Very good point. Thank you. We have added “or differences in macrophage activation status” to the sentence.
- Methods:
- Section 2.2: "After 2, 6, 12 and 24 h of incubation..." -> "After 2, 6, 12, and 24 h of incubation..." (Add comma).
Added. Thank you.
- Section 2.8: There are two sections labeled "2.8". The second one (Gene Expression) should be "2.9". All subsequent numbering in the Methods should be updated accordingly.
We appreciate your thorough review. We fixed the numbering of all sections.
- Section 2.9 (Statistical Analysis): Specify that data are presented as mean ± SEM.
Revised. Thanks
- Results:
- Figure 3 Legend: It says "five replicates," but the text says the assay was performed in "technical duplicates." Clarify the number of biological replicates.
We apologize for the confusion. We included the number of biological replicates in all legends
- Figure 5: The Y-axis label "Fluorescence Intensity" is clear, but a note explaining that higher fluorescence indicates higher phagocytosis would be helpful for a broader audience.
As per the reviewer’s suggestion, we have added the following statement in the figure legend: “The higher fluorescence indicates higher phagocytosis.”
- Section 3.2.5: When describing Figure 7, be precise. E.g., "a significant decrease was observed... at 6 hours post-treatment... at 10 MOI, whereas a significant increase... was observed at 12 hours... at 50 MOI."
Thank you. We fixed it as indicated.
- Discussion:
- Line 458-459: "...the immune system can often partially control infection." Consider rephrasing for accuracy, as the infection is often self-limiting, but the immune system doesn't necessarily "control" it before symptoms arise. Suggest: "...this may contribute to the eventual control of the infection in self-limiting cases."
We have added the following sentence: “This could explain the eventual control of the infection in self-limiting cases.”
- Line 480-482: "However, this finding should be interpreted with caution, as the lack of a statistically significant enhancement in bacterial killing may reflect limited assay sensitivity or a minimal biological effect." This is good, objective writing. Keep it.
Thank you.
- Formatting and Typos:
- Line 2: "Cam- 2 pylobacter" -> "Campylobacter"
This is not a typo; it follows the template style.
- Line 10: "Animal and Veterinary Sci- 10 ences" -> "Sciences"
This is not a typo; it follows the template style.
- Line 27: "C. jejunishowed" -> "C. jejuni showed"
Fixed
- Keywords (Line 36): Capitalize nouns as per journal style (typically all nouns are capitalized): "Macrophage; Epithelial Cell; Intestine; Immunity; Campylobacter; Communication; Crosstalk."
Fixed. Thank you
Reviewer 3 Report (Previous Reviewer 2)
Comments and Suggestions for Authors
Dear Authors,
The article “Epithelial-macrophage crosstalk in host responses to Campylobacter infection in humans” by Abdelaziz K. has been revised by the authors. However, it still requires necessary corrections before it can be accepted for further publication.
Below are my suggestions.
Abstract - 1-2 introductory sentences should be added at the beginning, answering the question of why the authors conducted this type of research. Why did they choose this particular topic? In addition, the results are too extensive and should be shortened. Line 28 - no space after “C. jejuni”
Introduction - important information on epidemiology is still missing. Why are Campylobacter infections so dangerous? The authors do not mention infectious doses or complications. Presenting campylobacteriosis from the point of view of a disease entity is very important in order to emphasize the importance of the research conducted by the authors.
Materials and methods – line 112 – please add a microscope photo (to Supplementary) of HT-29 culture after 4 hours to confirm that 90% of cells were indeed adhered.
Furthermore, I still believe that the incubation time with the strain was too short. In order to defend their position, the authors should discuss this issue in a separate paragraph in the Discussion section. Since Campylobacter symptoms appear on average after 3-5 days, this time is of great importance for the results obtained. The authors should refer to the publication: DOI: 10.3390/ijms262110634 , which clearly indicates that a 24-hour period does not have a statistically significant effect on changes occurring in the intestines. Here, it is important to point out that the dominance of Campylobacter over the cell line prevented adequate incubation time. Please devote a separate paragraph to this.
Once these important corrections have been made, the article will be ready to proceed to the next stages of publication.
Kind regards
Author Response
We sincerely appreciate the positive feedback from the reviewers and their valuable comments. Below, we provide a point-by-point response to each of their concerns.
Abstract - 1-2 introductory sentences should be added at the beginning, answering the question of why the authors conducted this type of research. Why did they choose this particular topic?
Thank you for your valuable suggestion. We have revised the statement to the following: ”Interactions between Campylobacter jejuni and host immune cells have been studied using various single-cell line models, such as macrophages or intestinal epithelial cells; however, these single-cell approaches do not fully capture the complexity of the host response. Investigating the interactions between these cell types offers a more comprehensive model for understanding Campylobacter–host dynamics.”
In addition, the results are too extensive and should be shortened.
We have shortened the results.
Line 28 - no space after “C. jejuni”
We added a space after C. jejuni.
Introduction - important information on epidemiology is still missing. Why are Campylobacter infections so dangerous? The authors do not mention infectious doses or complications. Presenting campylobacteriosis from the point of view of a disease entity is very important in order to emphasize the importance of the research conducted by the authors.
Thank you for your valuable suggestion. We have added a sentence detailing the infection rate, infectious dose, and the associated healthcare costs of this disease : “It is primarily transmitted through the consumption of contaminated animal products containing a low infectious dose of approximately 35 colony-forming units, causing an estimated 1.5 million cases of campylobacteriosis annually in the US, with associated healthcare costs ranging from $2 to $11 billion.”
Materials and methods – line 112 – please add a microscope photo (to Supplementary) of HT-29 culture after 4 hours to confirm that 90% of cells were indeed adhered.
We added a microscope photo to the supplementary file.
Furthermore, I still believe that the incubation time with the strain was too short. In order to defend their position, the authors should discuss this issue in a separate paragraph in the Discussion section. Since Campylobacter symptoms appear on average after 3-5 days, this time is of great importance for the results obtained. The authors should refer to the publication: DOI: 10.3390/ijms262110634 , which clearly indicates that a 24-hour period does not have a statistically significant effect on changes occurring in the intestines. Here, it is important to point out that the dominance of Campylobacter over the cell line prevented adequate incubation time. Please devote a separate paragraph to this.
Once these important corrections have been made, the article will be ready to proceed to the next stages of publication.
Thank you for your valuable suggestion. We have added the following paragraph to the discussion
Another important consideration is the relatively short incubation periods in our study, where IECs were exposed to Campylobacter for a maximum of 12 hours, which may not have been sufficient for the cells to fully release their modulatory molecules. Similarly, short-term exposure of HT-29 cells to supernatants from infected cells may not accurately capture the longer-term interactions that typically occur over a few days in an infected host. These observations are consistent with those of Anna Duda-Madej et al. (2025), who reported that a 24-hour exposure to berberine was insufficient to confer a protective effect on colonocytes exposed to C. jejuni post-culture supernatant, with statistically significant differences observed only after 72 hours of incubation.
We appreciate any further input from the reviwer.
Round 2
Reviewer 1 Report (Previous Reviewer 4)
Comments and Suggestions for Authors
Accepted.
Reviewer 2 Report (Previous Reviewer 3)
Comments and Suggestions for Authors
Dear Authors,
Thank you for submitting the revised version of your manuscript. We have reviewed the changes, and we are satisfied with the modifications made. The manuscript has been significantly improved and adequately addresses the points raised during the review.
Reviewer 3 Report (Previous Reviewer 2)
Comments and Suggestions for Authors
I appreciate that the authors have revised the manuscript, and I would now recommend it for publication.
This manuscript is a resubmission of an earlier submission. The following is a list of the peer review reports and author responses from that submission.
Round 1
Reviewer 1 Report
Comments and Suggestions for Authors
This manuscript explores epithelial–macrophage crosstalk in the host immune response to Campylobacter jejuni infection using an in vitro co-culture approach involving HT-29 intestinal epithelial cells and RAW264.7 macrophages. The use of epithelial cell–macrophage interaction models add physiological relevance compared to single-cell-type assays. The study integrates multiple functional assays: nitric oxide production, migration, phagocytosis, killing activity, and cytokine gene expression. The choice of multiplicities of infection (MOI 10 and 50) provides a comparative infection intensity model. However, while the experimental framework is conceptually sound, several aspects of the experimental design, data interpretation, and manuscript presentation require improvement.
Abstract
Line 15-20,the methodological sentence in the abstract is overly complex and contains several grammatical, structural, and conceptual issues that obscure its meaning. The sentence tries to convey multiple experimental elements—cell type, infection conditions, macrophage responses, and measured parameters—all in a single clause. The nested parentheses and brackets make the logic difficult to follow. It is not immediately clear whether nitric oxide (NO) production is the main indicator of macrophage activity or just one of several parameters assessed. The structure “macrophage activity [nitric oxide (NO) production], including migration, phagocytosis…” is logically inconsistent. NO production, migration, and phagocytosis are distinct functional parameters, not hierarchical subsets of “activity.” The phrase should be rewritten to parallel these endpoints properly. The segment “induced cytokine [interleukin (IL)-6, IL-1β, tumor necrosis factor (TNF)-α] and chemokine C-C motif chemokine ligand 2 (CCL2)” is grammatically incorrect and unclear. It should read “the expression of cytokines (IL-6, IL-1β, TNF-α) and the chemokine CCL2.”
Methods and Results
- No confirmation of bacterial removal. The authors filtered supernatants through 0.45 µm filters, but small OMVs or bacterial remnants could still pass through. The possibility that bacterial components rather than epithelial secretions caused macrophage activation is not ruled out.
- There is no mention of LPS or IFN-γ positive controls for macrophage activation. Including them would validate that the RAW264.7 cells respond appropriately.
- The manuscript claims that epithelial-derived soluble mediators activate macrophages, but no neutralization or inhibition experiments (e.g., IFN-γ blocking) were performed to identify the key signaling molecules involved. It is recommended to include an IFN-γ neutralization assay or cytokine profiling to pinpoint which mediators are responsible for macrophage activation.
- Line 129-130,It reads as though RAW264.7 cells were directly co-cultured with Campylobacter-infected HT-29 cells, while later context indicates that macrophages were actually treated with supernatants from infected HT-29 cells.
- Line 212,the conventional notation is 2^(-ΔΔCt) (or ), not 2ΔΔCT. Please correct the formatting to avoid confusion.
- Line 226, it is “HT-29 cells”.
Figures
- Figure numbers and titles should appear only in the figure captions, not embedded within the figure images themselves. Remove the “Figure X” text from within each figure panel (e.g., remove “Figure 1” and “Cytotoxicity” from the image of cytotoxicity assay).
- Figure1, caption incorrectly refers to the Griess assay, while the assay used was LDH cytotoxicity. And the authors present a “Positive control” group; however, the Materials and Methods section (Section 2.3), does not describe what was used as the positive control for the LDH cytotoxicity assay.
- Figure4, figures lack scale bars in microscopy images 4a-4c. The 4d panel appears to have been disproportionately resized, resulting in distorted aspect ratios and compressed axes. Additionally, the border around panel d should be removed.
- Figure7, the reported values of relative gene expression for all cytokines and chemokines are unusually low—mostly below 1 (ranging from 0.0x to 0.x). This pattern is biologically implausible, particularly since all targets show similarly small values, suggesting that the results may have been calculated or presented incorrectly. The authors should review their data-processing pipeline, correct any errors, and resubmit Figure 7 with recalculated and biologically interpretable fold-change values.
Conclusion
- The main conclusion—that Campylobacter evades killing despite macrophage activation—is plausible but insufficiently supported without intracellular bacterial survival imaging or CFU quantification showing viability within macrophages.
- The authors should temper statements such as “These findings expand our understanding of how Campylobacter can evade immune killing” to reflect correlative, not causal, evidence.
Author Response
We are grateful for the reviewers’ insightful feedback and have incorporated their suggestions into the revised manuscript. The abstract has also been updated for greater clarity. Your comments and recommendations have greatly improved the quality of our work.
Methods and Results
- No confirmation of bacterial removal. The authors filtered supernatants through 0.45 µm filters, but small OMVs or bacterial remnants could still pass through. The possibility that bacterial components rather than epithelial secretions caused macrophage activation is not ruled out.
We agree with the reviewer. To exclude this possibility, we included a control in which 10 MOI and 50 MOI of Campylobacter were incubated in DMEM without HT-29 cells and the resulting supernatant was then applied to RAW macrophages. This ensured that any observed macrophage activation was not due to direct exposure to bacterial components but rather to epithelial cell–derived factors. This clarification has been incorporated into the Materials and Methods and the Results sections, where we report: “While the Campylobacter supernatant from medium without cells significantly reduced NO production at both concentrations, significant NO production was observed in response to the supernatant from 10 MOI–treated HT-29 cells at 2- and 12-hour post-treatment.” These contrasting effects strongly support that the observed macrophage activation is driven by epithelial-derived mediators rather than by Campylobacter secretions alone.
- There is no mention of LPS or IFN-γ positive controls for macrophage activation. Including them would validate that the RAW264.7 cells respond appropriately.
Indeed, LPS was used in our prior trial as a control to help determine whether heat-killed, lysed, or live bacteria should be used. Live bacteria were selected because they induced a higher level of NO production, whereas heat-killed and lysed bacteria resulted in comparatively weaker NO responses. Regarding IFN-γ, positive controls were included in the kit used to measure IFN-γ production by stimulated HT-29 cells. We did not apply this control to macrophages, as various other secretory molecules could also contribute to macrophage activation, making IFN-γ a less specific indicator in this context. Our future studies will include mass spectrometry analyses to identify the specific soluble factors responsible for macrophage activation.
- The manuscript claims that epithelial-derived soluble mediators activate macrophages, but no neutralization or inhibition experiments (e.g., IFN-γ blocking) were performed to identify the key signaling molecules involved. It is recommended to include an IFN-γ neutralization assay or cytokine profiling to pinpoint which mediators are responsible for macrophage activation.
Thank you for this valuable suggestion. As this is a proof-of-concept study, our primary objective was to demonstrate the existence of crosstalk between Campylobacter-stimulated intestinal epithelial cells and macrophages, rather than to fully delineate the specific mediators involved. Identifying the exact epithelial-derived soluble factors that drive macrophage activation is indeed an important next step. We fully agree that neutralization assays (e.g., IFN-γ blocking) and/or comprehensive cytokine profiling would provide mechanistic insights into the key signaling molecules responsible for these responses.
We have clarified this in the revised Discussion and have highlighted that ”Future studies will include mass spectrometry–based profiling of the secreted molecules in the supernatants from Campylobacter-infected HT-29 cells, along with targeted cytokine-neutralization assays in macrophages, to identify the specific mediators responsible for macrophage activation.”
- Line 129-130,It reads as though RAW264.7 cells were directly co-cultured with Campylobacter-infected HT-29 cells, while later context indicates that macrophages were actually treated with supernatants from infected HT-29 cells.
Thank you for your thorough review and we apologize for the confusion. The sentence has been revised to the following: “NO production in RAW264.7 cells (monocyte/macrophage-like cells), following treatment with the secretory products of Campylobacter-infected HT-29 cells, was quantified using the Griess assay.”
- Line 212,the conventional notation is 2^(-ΔΔCt) (or ), not 2ΔΔCT. Please correct the formatting to avoid confusion.
Corrected. Thank you.
- Line 226, it is “HT-29 cells”.
- Thank you.
Figures
- Figure numbers and titles should appear only in the figure captions, not embedded within the figure images themselves. Remove the “Figure X” text from within each figure panel (e.g., remove “Figure 1” and “Cytotoxicity” from the image of cytotoxicity assay).
We completely agree. The MDPI team offered free formatting, and we have encountered several formatting issues in the submitted version. We sincerely apologize for taking up your valuable time to address these problems.
- Figure1, caption incorrectly refers to the Griess assay, while the assay used was LDH cytotoxicity. And the authors present a “Positive control” group; however, the Materials and Methods section (Section 2.3), does not describe what was used as the positive control for the LDH cytotoxicity assay.
We apologize for this mistake and thank the reviewers for their thorough review. We used the colorimetric CyQUANT™ LDH cytotoxicity assay kit (Thermo Fisher Scientific, Greenville County, SC, USA)
We revised our statement to the following “The Colorimetric CyQUANT™ LDH Cytotoxicity Assay Kit (Thermo Fisher Scientific, Greenville County, SC, USA) was used to evaluate the effects of two concentrations of C. jejuni on the viability of HT-29 cells.”
The positive control used was the LDH Positive Control, which serves as an internal control included in the kit.
- Figure4, figures lack scale bars in microscopy images 4a-4c. The 4d panel appears to have been disproportionately resized, resulting in distorted aspect ratios and compressed axes. Additionally, the borderaround panel d should be
We thank the reviewer for this observation. Unfortunately, the microscopy images were captured using an older camera available in our laboratory that does not provide automatic scale bars. The images presented are representative of those captured, and all images were taken at the same magnification. The border around panel 4d has also been removed as suggested.
- Figure7, the reported values of relative gene expression for all cytokines and chemokines are unusually low—mostly below 1 (ranging from 0.0x to 0.x). This pattern is biologically implausible, particularly since all targets show similarly small values, suggesting that the results may have been calculated or presented incorrectly. The authors should review their data-processing pipeline, correct any errors, and resubmit Figure 7 with recalculated and biologically interpretable fold-change values.
- We sincerely thank the reviewer for this valuable comment. We would like to clarify that the relative expression values in Figure 7 were calculated based on normalization to the reference gene β-actin, which showed stable Ct values across all treatment conditions. Because the data were not normalized to a calibrator set to “1,” some values appear below 1. This reflects true biological responses relative to β-actin rather than an error in data processing. A similar expression pattern was also observed in our previous published work, particularly for genes with modest induction levels.
- Importantly, our primary objective was to determine whether secreted mediators from Campylobacter-stimulated HT-29 cells induced statistically significant changes in gene expression compared to the negative control. In response to the reviewer’s suggestion, we have now re-plotted the data using a fold-change presentation. Nonetheless, we believe relative expression values remain biologically meaningful since no treatment comparisons were performed in this study.
Conclusion
- The main conclusion—that Campylobacter evades killing despite macrophage activation—is plausible but insufficiently supported without intracellular bacterial survival imaging or CFU quantification showing viability within macrophages.
We would like to clarify that the percentage killing was calculated based on the quantification of Campylobacter CFUs recovered from RAW cells. For clarity, the killing percentage has been indicated above each column in the figure.
- The authors should temper statements such as “These findings expand our understanding of how Campylobacter can evade immune killing” to reflect correlative, not causal, evidence.
We have revised the conclusion statement to the following: “While these findings provide supporting evidence that Campylobacter can evade immune killing despite immune system activation, further studies are needed to validate this observation and to elucidate the mechanisms by which Campylobacter survives within activated macrophages.”
Reviewer 2 Report
Comments and Suggestions for Authors
Dear Authors,
I have serious concerns about the correctness of the methods used in the Manuscript.
First of all, the abstract is written incorrectly. The results take up too much space. Furthermore, the first two sentences refer to the same thing. In addition, the authors should mention here which strain the study concerns. By writing generally about Campylobacter, they mislead the reader, as we later learn that the study focuses on C. jejuni. The wording used suggests that the authors tested several species.
The introduction also does not address the importance of infection with this microorganism. Why did it become the subject of research? Yes, we read about the infection process, what happens, but it is not described from the point of view of its effects on the human body.
Methodology: this is where I have the most concerns. 1. Why did the authors not give the HT-29 cells more time to adhere? After all, only a small proportion adhered after this time. So what was the point of the 4 hours? Couldn't the cells with bacteria have been added immediately? In my opinion, the subsequent steps were therefore performed incorrectly. The authors suggest that they worked on the supernatant from the contact, but it was not the correct supernatant, as HT-29 was not properly presented. 2. Why was the Campylobacter-HT-29 contact only 12 hours? Why did the authors not examine several time points? It is well known that in the case of Campylobacter, the longer the contact, the more virulence factors the bacteria have at their disposal, hence the time-related symptoms of campylobacteriosis.
These two points strongly affect the incorrectness of the other methods used. I believe that the authors would have achieved a much better result with supernatants from contacts using membranes.
Therefore, I believe that it is not suitable for publication in its current form.
Author Response
We are grateful for the reviewers’ insightful feedback and have incorporated their suggestions into the revised manuscript. The abstract has also been updated for greater clarity. Your comments and recommendations have greatly improved the quality of our work.
The introduction also does not address the importance of infection with this microorganism. Why did it become the subject of research? Yes, we read about the infection process, what happens, but it is not described from the point of view of its effects on the human body.
As per the reviewer's suggestion, we have included the following statement at the beginning of the introduction: ”Campylobacteriosis in healthy individuals is typically a self-limiting disease; however, Campylobacter infections in immunocompromised and elderly individuals can lead to severe clinical manifestations, necessitating antimicrobial treatment [5]. To date, there is a limited understanding of host immune surveillance in response to this pathogen and its ability to cause disease, which has hindered the development of innovative strategies for its prevention and treatment in humans.”
Methodology: this is where I have the most concerns. 1. Why did the authors not give the HT-29 cells more time to adhere? After all, only a small proportion adhered after this time. So what was the point of the 4 hours? Couldn't the cells with bacteria have been added immediately? In my opinion, the subsequent steps were therefore performed incorrectly.
We thank the reviewer for raising this important point. Optimizing the methodology required over a year of careful adjustments, including determining the appropriate stimulant (heat-killed, lysed, or live Campylobacter), and bacterial concentration (MOI) that could elicit a measurable response without overgrowing. We also evaluated the optimal duration for exposing macrophages to bacteria to ensure activation without compromising macrophage viability, and studied RAW cell growth dynamics to determine the appropriate number of cells to seed and the necessary adherence time.
Through these optimizations, we determined that a 4-hour adherence period allowed the majority of RAW cells to attach fully to the tissue culture plates. Extending this time further would have led to cell division, complicating accurate MOI calculations for Campylobacter. While adding the bacteria immediately after seeding was considered, we were concerned that rapid bacterial multiplication could interfere with macrophage viability and adherence. Based on our optimization studies, live Campylobacter induced the most robust responses in cultured macrophages under these conditions.
Thus, the 4-hour adherence period represents a carefully determined compromise that ensures both reliable macrophage activation and reproducibility of the MOI.
The authors suggest that they worked on the supernatant from the contact, but it was not the correct supernatant, as HT-29 was not properly presented. 2. Why was the Campylobacter-HT-29 contact only 12 hours? Why did the authors not examine several time points? It is well known that in the case of Campylobacter, the longer the contact, the more virulence factors the bacteria have at their disposal, hence the time-related symptoms of campylobacteriosis.
We thank the reviewer for this observation. In our optimization trials, we indeed extended the timepoint to 24 hours, as we agree that Campylobacter releases more virulence factors over extended periods, and our prior studies showed that LPS stimulation of RAW cells resulted in higher nitric oxide production at later time points (2, 6, and 12 hours). However, we observed that prolonged exposure led to bacterial overgrowth, which completely destroyed the HT-29 cells. Additionally, the media turned yellow, indicating changes in pH due to bacterial metabolism, further compromising cell viability.
Based on these observations, we chose to limit the incubation to 12 hours to assess subsequent parameters. This timepoint was sufficient to detect alterations in the immune responses of both HT-29 cells and macrophages stimulated with HT-29 supernatants. While we acknowledge that longer exposure could elicit stronger responses, the 12-hour timepoint provided more reliable and reproducible outcomes without compromising cell viability.
These two points strongly affect the incorrectness of the other methods used. I believe that the authors would have achieved a much better result with supernatants from contacts using membranes.
Therefore, I believe that it is not suitable for publication in its current form.
We sincerely thank the reviewer for their valuable feedback and insightful comments. Your guidance has greatly enhanced the quality of our work, and we truly appreciate the time and effort you have dedicated.
Reviewer 3 Report
Comments and Suggestions for Authors
The manuscript addresses a pertinent and understudied aspect of Campylobacter pathogenesis: the crosstalk between intestinal epithelial cells (IECs) and macrophages. The experimental approach is sound and the topic is well-suited for Microorganisms. The central finding—that epithelial-derived signals activate macrophages but fail to confer bactericidal activity—is novel and significant. However, the manuscript in its current form requires substantial revision to strengthen the data interpretation, clarify methodological details, and improve the overall narrative flow and rigor. The conclusions are promising but are not yet fully supported by the presented data without further clarification and contextualization.
Major Points
-
Lack of Protein-Level Confirmation for Cytokine/Chemokine Data:
A major weakness is the exclusive reliance on qRT-PCR for measuring key cytokines (IL-6, IL-1β, TNF-α) and chemokine (CCL2). Gene expression data does not necessarily correlate with protein secretion or bioactivity. The entire narrative about the immunomodulatory role of the supernatants is built on mRNA levels alone.
The authors must include protein-level data (e.g., ELISA) for at least the most critical cytokines (e.g., IL-6, CCL2) to confirm that the observed gene expression translates to actual protein secretion. This is critical for validating the central hypothesis.
-
Incomplete and Potentially Misleading Cytotoxicity Assay:
Figure 1 and its description are problematic. The legend states cytotoxicity was measured by the "Griess assay" (which measures NO), but the method and Y-axis label correctly identify it as an LDH assay. More importantly, the positive control is not described, and the data shows no significant cytotoxicity for the infected groups. However, without a clear positive control (e.g., lysed cells) demonstrating what 100% cytotoxicity looks like in this assay, it is difficult to assess the assay's sensitivity and truly confirm the absence of cytotoxic effects.
Correct the figure legend. Clearly describe and include the data for the positive control in Figure 1. A statement confirming that the positive control yielded the expected high cytotoxicity should be added to the results section.
-
Overstated Conclusions from the Killing Assay:
The result "no significant killing" (Fig. 6) is presented as a key finding. However, the lack of statistical significance could be due to high variability and a potentially underpowered assay (n=6 might be insufficient for this type of highly variable biological assay). The authors should show the raw CFU data for T0 and T2 to allow the reader to better assess the biological trend.
Present the raw CFU data (T0 and T2) in a supplementary figure or table. The discussion should be tempered to acknowledge that while there was no statistical enhancement in killing, the assay may not have been sensitive enough to detect a biological effect, or that the effect is indeed negligible.
-
Clarification of the "Campylobacter Metabolites" Control:
The term "Campylobacter metabolites" is used in the abstract and text, but the method describes it as supernatant from Campylobacter incubated in DMEM without cells. This is more accurately a "bacterial supernatant" containing secreted factors and possibly shed membrane components, not purified metabolites. This distinction is important.
Use a more precise term throughout the manuscript, such as "cell-free Campylobacter supernatant" or "soluble bacterial factors." Clearly state in the methods what components this supernatant likely contains (e.g., secreted proteins, OMVs, etc.).
-
Minor Points (Recommended Revisions)
-
Introduction:
-
Improve the flow by better connecting the known roles of IECs and macrophages. The last paragraph of the introduction is excellent and clearly states the knowledge gap and study aim.
-
-
Methods:
-
Section 2.2: Specify the volume of DMEM in which C. jejuni was incubated alone for the control supernatant.
-
Section 2.8.1: The title and text of this section refer to RNA isolation from HT-29 cells, but the subsequent qRT-PCR is performed on macrophages. This is confusing. The title should be "RNA Isolation and cDNA Preparation from Macrophages" or similar.
-
Statistical Analysis: Clarify the "n" for each experiment. Is n=6 biological replicates (independent cell cultures) or technical replicates? Biological replicates are essential for the conclusions drawn.
-
-
Results:
-
Figure 3: The finding that 50 MOI supernatant suppresses NO production is intriguing. The discussion would be strengthened by speculating on potential mechanisms (e.g., higher load of a specific bacterial immunosuppressive factor in the IEC supernatant).
-
Figure 4: The result states that both 10 and 50 MOI supernatants "significantly altered" phagocytosis. This is vague. The text should explicitly state that 10 MOI increased it while 50 MOI decreased it at 12h, emphasizing the dose-dependent effect.
-
Figure 7: The patterns of cytokine expression are complex and time-dependent. The discussion should more deeply interpret these temporal changes rather than just stating they were "altered."
-
-
Discussion:
-
The discussion is generally good but could be more focused. It should first directly answer the questions posed in the introduction regarding migration, phagocytosis, killing, and cytokine response, before broadening out to the wider implications.
-
The speculation that infectious dose may explain individual variation is interesting but should be presented more cautiously, as the in vitro model does not fully recapitulate the in vivo dose-response.
-
-
Formatting and Typos:
-
The author affiliations are inconsistent (US vs. USA).
-
There is a typo in the results title "3.1.2. Campylobacter enhanced IFN-γ production by IECs" (check for "enhanced").
-
Ensure all figure citations in the text are correct (e.g., the text for Fig. 4 mentions 2, 6, and 12 hours, but the figure itself may not show all time points).
-
-
Author Response
Major Points
- Lack of Protein-Level Confirmation for Cytokine/Chemokine Data:
A major weakness is the exclusive reliance on qRT-PCR for measuring key cytokines (IL-6, IL-1β, TNF-α) and chemokine (CCL2). Gene expression data does not necessarily correlate with protein secretion or bioactivity. The entire narrative about the immunomodulatory role of the supernatants is built on mRNA levels alone.
The authors must include protein-level data (e.g., ELISA) for at least the most critical cytokines (e.g., IL-6, CCL2) to confirm that the observed gene expression translates to actual protein secretion. This is critical for validating the central hypothesis.
We thank the reviewer for highlighting the importance of evaluating cytokine responses at the protein level. We fully agree that mRNA expression does not always equate to protein secretion or biological activity. In this study, our primary objective was to establish the initial immunomodulatory effects of Campylobacter-stimulated HT-29 supernatants on macrophage activation using a validated transcriptional readout. The selected panel of cytokines and chemokines represents well-established early immune markers that reliably correlate with downstream functional responses in this model system.
We have now clarified this rationale in the manuscript and expanded the Discussion to acknowledge that future work will incorporate protein-level analyses such as ELISA to capture secreted mediators and confirm the magnitude of cytokine responses. These studies are planned as part of our broader research effort to comprehensively characterize both gene and protein expression dynamics following Campylobacter exposure.
To acknowledge this limitation, we have added the following statement before the conclusions: “Future studies will include mass spectrometry–based profiling of the secreted molecules in the supernatants from Campylobacter-infected HT-29 cells, along with targeted cytokine-neutralization assays in macrophages, to identify the specific mediators responsible for macrophage activation. Additionally, incorporating protein-level analyses, such as ELISA, will enable quantification of secreted cytokines and confirmation of the magnitude of the observed immune responses.”
- Incomplete and Potentially Misleading Cytotoxicity Assay:
Figure 1 and its description are problematic. The legend states cytotoxicity was measured by the "Griess assay" (which measures NO), but the method and Y-axis label correctly identify it as an LDH assay. More importantly, the positive control is not described, and the data shows no significant cytotoxicity for the infected groups. However, without a clear positive control (e.g., lysed cells) demonstrating what 100% cytotoxicity looks like in this assay, it is difficult to assess the assay's sensitivity and truly confirm the absence of cytotoxic effects.
Correct the figure legend. Clearly describe and include the data for the positive control in Figure 1. A statement confirming that the positive control yielded the expected high cytotoxicity should be added to the results section.
We apologize for this mistake and thank the reviewers for their thorough review. We used the colorimetric CyQUANT™ LDH cytotoxicity assay kit (Thermo Fisher Scientific, Greenville County, SC, USA).
We revised our statement in the M&M section to the following “The Colorimetric CyQUANT™ LDH Cytotoxicity Assay Kit (Thermo Fisher Scientific, Greenville County, SC, USA) was used to evaluate the effects of two concentrations of C. jejuni on the viability of HT-29 cells. Briefly, cells were seeded in five replicates as described above and incubated with live Campylobacter at an MOI of 10 or 50”
We revised our statement in the results section to the following “This assay was performed to assess whether C. jejuni at two different concentrations (10 and 50 MOI) exerts cytotoxic effects on HT-29 cells using a lactate dehydrogenase (LDH) release assay.”
The positive control used was the LDH Positive Control, which serves as an internal control included in the kit. We have added the following statement to M&M “LDH supplied in the kit was used as the positive control, in accordance with the manufacturer’s instructions.”
Regarding the use of a positive control in the LDH assay, we followed the manufacturer’s recommended procedure for inducing spontaneous LDH release. Specifically, 10 µL of sterile water was added to the designated wells containing cells (triplicate), followed by gentle mixing. The resulting spontaneous LDH release values were then used in the calculation of percent cytotoxicity according to the formula provided in the kit instructions.
Overstated Conclusions from the Killing Assay:
The result "no significant killing" (Fig. 6) is presented as a key finding. However, the lack of statistical significance could be due to high variability and a potentially underpowered assay (n=6 might be insufficient for this type of highly variable biological assay). The authors should show the raw CFU data for T0 and T2 to allow the reader to better assess the biological trend.
Present the raw CFU data (T0 and T2) in a supplementary figure or table. The discussion should be tempered to acknowledge that while there was no statistical enhancement in killing, the assay may not have been sensitive enough to detect a biological effect, or that the effect is indeed negligible.
As per the reviewer’s suggestions, we have provided a supplementary table for the Rawa CFU data and added the following statement to the discussion: “However, this finding should be interpreted with caution, as the lack of a statistically significant enhancement in bacterial killing may reflect limited assay sensitivity or a minimal biological effect.”
- Clarification of the "Campylobacter Metabolites" Control:
The term "Campylobacter metabolites" is used in the abstract and text, but the method describes it as supernatant from Campylobacter incubated in DMEM without cells. This is more accurately a "bacterial supernatant" containing secreted factors and possibly shed membrane components, not purified metabolites. This distinction is important.
Use a more precise term throughout the manuscript, such as "cell-free Campylobacter supernatant" or "soluble bacterial factors." Clearly state in the methods what components this supernatant likely contains (e.g., secreted proteins, OMVs, etc.).
We followed the reviewer’s guidance and replaced the term “metabolites” with “cell-free Campylobacter supernatants.” We also revised the description under the Nitric Oxide subsection in the Results section to the following: ”While the Campylobacter cell-free supernatant from medium without HT-29 cells—likely containing secreted proteins, peptides, metabolic by-products, OMVs, and lipooligosaccharides (LOS)—significantly reduced NO production at both concentrations, significant NO production was observed in response to the supernatant from 10 MOI–treated HT-29 cells at 2- and 12-hour post-treatment. “
- Minor Points (Recommended Revisions)
- Introduction:
- Improve the flow by better connecting the known roles of IECs and macrophages. The last paragraph of the introduction is excellent and clearly states the knowledge gap and study aim.
- Introduction:
We have added the following paragraph to link IECS and macrophages: ”In addition to examining Campylobacter–macrophage interactions, studies using IECs have delineated several virulence factors associated with Campylobacter motility, adhesion, and invasion, as well as the innate immune responses initiated upon bacterial entry into the intestinal tract. While investigations focused on a single innate immune cell type—such as IECs or macrophages—have provided important insights into Campylobacter virulence and host responses, they do not fully capture the complexity of cellular interactions during infection.”
- Methods:
- Section 2.2:Specify the volume of DMEM in which jejuni was incubated alone for the control supernatant.
We have added the following paragraph under section 2.5 “Briefly, RAW264.7 cells were seeded in five replicates at a density of 4 × 10⁵ cells/well in 500 μL of supplemented DMEM in 48-well plates and incubated for 3 h at 37 °C in a humidified 5% CO₂ incubator to allow adherence. After 3 h, the DMEM medium was replaced with supernatants from Campylobacter-treated HT-29 cells or cell-free Campylobacter supernatants, and cells were incubated for 2-, 6-, or 12-hours under the same conditions. The supernatants were then collected for measuring NO.”
- Section 2.8.1:The title and text of this section refer to RNA isolation from HT-29 cells, but the subsequent qRT-PCR is performed on macrophages. This is confusing. The title should be "RNA Isolation and cDNA Preparation from Macrophages" or similar.
Thank you: we have added the word “macrophages” to the title.
- Statistical Analysis:Clarify the "n" for each experiment. Is n=6 biological replicates (independent cell cultures) or technical replicates? Biological replicates are essential for the conclusions drawn.
We have revised our statement under statistical analysis to the following: “Data from 5–6 biological replicates per assay were analyzed using one-way analysis of variance (ANOVA) to determine whether statistically significant differences existed among the treatment groups. When ANOVA indicated a significant effect, Tukey’s multiple comparison test was applied as a post-hoc analysis to identify which specific groups differed from each other.”
- Results:
- Figure 3:The finding that 50 MOI supernatant suppresses NO production is intriguing. The discussion would be strengthened by speculating on potential mechanisms (e.g., higher load of a specific bacterial immunosuppressive factor in the IEC supernatant).
Thank you for your suggestion. We have revised the relevant information under the discussion section “Moreover, the suppression of NO production and macrophage phagocytosis in cells treated with supernatants from 50 MOI–infected IECs is likely due to the presence of higher levels of specific bacterial immunosuppressive factors in the cell-free medium, which may account for dose-dependent variations in the immune system’s ability to control infection.”
- Figure 4:The result states that both 10 and 50 MOI supernatants "significantly altered" phagocytosis. This is vague. The text should explicitly state that 10 MOI increased it while 50 MOI decreased it at 12h, emphasizing the dose-dependent effect.
Revised as suggested: “Treatment of macrophages with supernatants from Campylobacter-infected IECs at 10 MOI significantly enhanced phagocytosis at 6 hours post-treatment, with no significant changes observed at 2 or 12 hours, while treatment with the supernatant from IECs infected at 50 MOI significantly reduced macrophage phagocytic activity at 12 hours post-treatment, emphasizing the dose-dependent effect.”
- Figure 7:The patterns of cytokine expression are complex and time-dependent. The discussion should more deeply interpret these temporal changes rather than just stating they were "altered."
Thanks for the suggestion: here is the revised paragraph “The secretory molecules from C. jejuni–infected HT-29 cells at both 10 and 50 MOI induced macrophage expression of pro-inflammatory cytokines—including IL-1β, IL-6, and TNF-α—as well as the chemokine CCL2, in a dose- and time-dependent manner, all of which have been previously reported to contribute to the initiation of inflammation and the development of gastroenteritis. Nevertheless, despite significant changes in gene expression, enhanced phagocytosis, and increased nitric oxide production, macrophages exhibited a limited ability to kill Campylobacter, highlighting the need to investigate additional molecular and cellular mechanisms underlying pathogen survival.
- Discussion:
- The discussion is generally good but could be more focused. It should first directly answer the questions posed in the introduction regarding migration, phagocytosis, killing, and cytokine response, before broadening out to the wider implications.
Thank you for your valuable suggestion. We have added the following statements to the discussion as per your suggestion:
“Although these in vitro studies provide valuable insights into host-microbe interactions, including the identification of some mechanistic pathways and characterization of immune responses in these cells during Campylobacter infection, they do not fully recapitulate the in vivo gut environment. For example, most experiments have directly exposed in vitro–cultured macrophages to Campylobacter.”
“Evaluating the interplay between Campylobacter and IECS, including the effects of their secreted molecules on macrophage activation, may offer deeper insight into immune activation than analyses focused on individual immune cell types.”
- The speculation that infectious dose may explain individual variation is interesting but should be presented more cautiously, as the in vitro model does not fully recapitulate the in vivo dose-response.
Revised as suggested. “Moreover, the suppression of NO production and macrophage phagocytosis in cells treated with supernatants from 50 MOI–infected IECs is likely due to the presence of higher levels of specific bacterial immunosuppressive factors in the cell-free medium, which may account for dose-dependent variations in the immune system’s ability to control infection [32]. However, since the in vitro model does not fully capture the complexities of the host response, further validation of these findings in a dose-response study in an animal infection model is warranted.”
- Formatting and Typos:
- The author affiliations are inconsistent (US vs. USA).
Corrected.
- There is a typo in the results title "3.1.2. Campylobacter enhanced IFN-γ production by IECs" (check for "enhanced").
Corrected.
- Ensure all figure citations in the text are correct (e.g., the text for Fig. 4 mentions 2, 6, and 12 hours, but the figure itself may not show all time points).
We greatly appreciate your thorough review. The incubation time was corrected to 24 hours.
Reviewer 4 Report
Comments and Suggestions for Authors
This study “Epithelial–Macrophage Crosstalk in Host Responses to Campylobacter Infection in Humans” presents the interplay between Campylobacter, intestinal epithelial cells and macrophages. This study examined whether soluble factors secreted from Campylobacter-infected HT-29 cells (human colorectal adenocarcinoma cells which express characteristics of mature intestinal cells) at 10 and 50 multiplicities of infection (MOI) influence RAW 264.7 macrophage activity [nitric oxide (NO) production], including migration, phagocytosis, bacterial killing, and induced cytokine [interleukin (IL)-6, IL-1β, tumor necrosis factor (TNF)-α] and chemokine C-C motif chemokine ligand 2 (CCL2). Overall, this research topic holds practical significance and demonstrates a degree of innovation. The experimental design is relatively systematic, and the data are fairly comprehensive. However, there are several areas for improvement:
- In order to increase the coherence of the full text, it is recommended to mark the serial number after each formula, such as “ 1-1”.
- In this paper, it is mentioned that the use of Image J software version 1.54p ( NIH, USA ) to manually quantify the migrated cells can explain the method of reducing the error and enhance the reliability of the article.
- There are 10 and 50 moles of Campylobacter mentioned in the article. Please comprehensively analyze the difference between the molar concentrations of Campylobacter 10 and 50 MoL on the interaction between intestinal epithelial cells and macrophages.
Author Response
We sincerely thank the reviewer for their valuable feedback and insightful comments. Your guidance has greatly enhanced the quality of our work, and we truly appreciate the time and effort you have dedicated.
1- In order to increase the coherence of the full text, it is recommended to mark the serial number after each formula, such as “ 1-1”.
We have replaced the original formula with the following, including a description for clarity:
% Cytotoxicity = ((Compound-treated LDH activity – Spontaneous LDH activity) / (Maximum LDH activity – Spontaneous LDH activity)) × 100
- Compound-treated LDH: LDH activity in the HT-29 supernatant-treated wells
- Spontaneous LDH: LDH activity in the distilled water-treated wells (baseline release)
- Maximum LDH: LDH activity after complete lysis of cells (total release)
2- In this paper, it is mentioned that the use of Image J software version 1.54p ( NIH, USA ) to manually quantify the migrated cells can explain the method of reducing the error and enhance the reliability of the article.
We have provided additional information to clarify that migrated cells were manually quantified using ImageJ software (version 1.54p). For each experimental condition, four images were captured from separate, non-overlapping fields of the membrane, and the number of migrated cells in each image was counted to ensure accuracy and reproducibility of the data.
We have revised our statement to the following: “Migrated cells were quantified in four images captured from different non-overlapping fields using ImageJ software (version 1.54p).”
3- There are 10 and 50 moles of Campylobacter mentioned in the article. Please comprehensively analyze the difference between the molar concentrations of Campylobacter 10 and 50 MoL on the interaction between intestinal epithelial cells and macrophages.
Thank you for your comment. However, we would like to clarify that HT-29 cells were stimulated using a multiplicity of infection (MOI) of 10 or 50, meaning that each cell was exposed to either 10 or 50 bacteria. This does not refer to molar concentration.
Round 2
Reviewer 2 Report
Comments and Suggestions for Authors
Dear Authors,
Thank you very much for your comprehensive explanations regarding my objections. I believe that if the authors add explanations to these doubts in the manuscript, the article can be reconsidered. However, in the version without this information, it misleads researchers. It is necessary to specify exactly what times were studied and what was observed during those times, and therefore, for further research, the ... time point was selected. In the case of the RAW 264.7 line, did the authors check adhesion after 12 hours? In my experience, this is the optimal time when proliferation has not yet occurred and the vast majority of cells are already adhered. This would avoid the error associated with a large number of cells not adhering. In addition, there are also cells that must differentiate after seeding. To determine the exact MOI, an additional well (or 3, to be sure) is set up on the plate, and after a specified time, their number is determined and the appropriate amount of bacteria is added to establish the desired MOI. Good luck.
Author Response
We sincerely thank the reviewer for considering our responses and providing us the opportunity to resubmit the revised manuscript.
We have included more details in section 2.2 in the M&M
“The cells were allowed to adhere for 4 h before being infected with C. jejuni at a multiplicity of infection (MOI) of 10 or 50 in 1 mL of fresh DMEM per well or maintained in DMEM without C. jejuni (negative control). C. jejuni was added to DMEM alone, without cells, at an MOI of 10 or 50. After 2, 6, 12 and 24 h of incubation at 37 °C in a humidified 5% CO₂ incubator. However, prolonged incubation for 24 hours resulted in excessive bacterial overgrowth, leading to complete destruction of the HT-29 cell monolayers. The culture media also turned yellow, indicating a drop in pH due to bacterial metabolism, which further compromised cell viability. Based on these observations, incubation times were limited to a maximum of 12 h. Following incubation, the plates were centrifuged at 300 × g for 3 minutes, and the supernatant collected at 12 h was subsequently filtered through a 0.45 μm syringe filter to remove bacterial cells.”
Additionally, in the results section (3.1.2), we have included the following statement: “Based on these observations, supernatants collected at the 12-hour time point were selected for subsequent experiments evaluating the ability of secreted bacterial molecules to activate macrophages.”
Regarding the adherence of RAW cells, it is well documented that approximately 90% of these cells adhere within 3 hours (see protocol: https://www.ubigene.us/application/raw2647-cell-differentiate.html). We assume the reviewer may be referring to THP-1 cells; as for RAW cells, adherence was not a concern because we did not infect the cells with Campylobacter, and therefore the exact number of seeded cells or the percentage adhered at the 3-hour mark did not impact the experiment. Even if 10% of the cells were not fully adhered at 3 hours, the 12-hour incubation period would allow sufficient time for the remaining cells to adhere.
In response to the reviewer’s suggestion, we conducted an additional experiment in which RAW cells were seeded in six replicates, incubated for 4 hours, the supernatant was removed, and the adherent cells were trypsinized and counted. As shown in the table below, the average adherence rate was 96%.
|
Replicates |
Number of cells seeded/1ml |
Number of cells adhered/1ml |
Adherence % |
|
1 |
1x10^6 |
8.9 ×10^5 |
89 |
|
2 |
1x10^6 |
9.3×10^5 |
93 |
|
3 |
1x10^6 |
1.02×10^6 |
100 |
|
4 |
1x10^6 |
1.02×10^6 |
100 |
|
5 |
1x10^6 |
9.6×10^5 |
96 |
|
6 |
1x10^6 |
9.9×10^5 |
99 |
We hope this addresses your concerns, and we are happy to provide any further clarification if needed.
Reviewer 3 Report
Comments and Suggestions for Authors
The revision work carried out by the authors is appreciable. I have no further comments to make.
Author Response
Thank you very much for helping us improve the quality and clarity of our manuscript through your valuable comments and suggestions.